

# The impact of mineral dust on the day-to-day variability of stratiform cloud glaciation occurrence

Diego Villanueva[1], Bernd Heinold[1], Patric Seifert[1], Hartwig Deneke[1], Martin Radenz[1], Ina Tegen[1]

[1]Leibniz Institute for Tropospheric Research, Leipzig, 04318, Germany

*Correspondence to*: Diego Villanueva (ortiz@tropos.de)

**Abstract.** Two different A-Train satellite cloud phase products were analysed together with an aerosol model reanalysis to assess the global day-to-day variability of cloud thermodynamic phase. This variability was analysed for different mixing-ratios of fine and coarse mineral dust during the period 2007-2010 and within a temperature range from +3°C to -42°C.

Night-time stratiform clouds were analysed, including stratocumulus, altocumulus, altostratus and cirrus clouds. This analysis showed that the phase of stratiform clouds is highly dependent on temperature and latitude. However, at equal temperature the average occurrence of fully glaciated stratiform clouds was found to increase for higher dust mixing-ratios on a day-to-day basis at mid- and high latitudes. At -15°C, the increment of ice cloud occurrence between the lowest and highest mixing-ratio was found to be higher for fine dust (+10 % to +18 % occurrence) than for coarse dust (+5 % to + 10%). Surprisingly, the

increments were higher in remote regions (e.g. southern high latitudes) where the average dust-mixing ratios are low.

## 1    Introduction

Aerosol-cloud interactions affect the Earth's climate through different mechanisms. These include impacts of aerosol particles on cloud glaciation that influence the cloud albedo, cloud lifetime and precipitation. Specifically, there is growing evidence towards the global role of mineral dust aerosol (or of ice nucleating particles correlated to dust aerosol) in heterogeneous cloud

ice formation (Boose et al., 2016; Kanitz et al., 2011; Seifert et al., 2010; Tan et al., 2014; Vergara-Temprado et al., 2017; Zhang et al., 2018). Cloud droplets can freeze heterogeneously between 0°C and -42°C after interacting with aerosols acting as Ice Nucleating Particles (INP) or already existing ice particles (Hoose and Möhler, 2012). It has been shown that specific aerosol types such as mineral dust and biogenic particles can act efficiently as INP already at temperatures between -10 and -20°C (Atkinson et al., 2013). Mineral dust aerosol is an efficient INP emitted from arid regions, mainly from

the Saharan and Asian deserts. Mineral dust is therefore suspected to be a main contributor to atmospheric INP, especially in the northern hemisphere (Vergara-Temprado et al., 2017). The mixing-ratio of dust aerosol in the northern hemisphere is typically one to two orders of magnitude larger than in the southern hemisphere (Vergara-Temprado et al., 2018). Despite of this, several dust sources exist at the southern mid-latitudes (e.g. Patagonia, South Africa and Australia) and simulations show that long-range transported dust, although sporadic, can result in considerable dust concentrations even in remote areas

(Johnson et al., 2011; Li et al., 2008; Vergara-Temprado et al., 2017).



The freezing efficiency of INPs depends mainly on their surface area concentration (Atkinson et al., 2013; Hartmann et al., 2016; Murray et al., 2011; Niedermeier et al., 2011, 2015; Price et al., 2018). While the number concentration of dust aerosol is generally dominated by fine (particle diameter < 0.5 µm) dust, the surface area concentration is often dominated by both fine and coarse (particle diameter > 1 µm) dust particles (Mahowald et al., 2014). Moreover, atmospheric lifetime of fine dust

is longer than that of coarse dust due to the lower dry deposition rates of finer particles (Mahowald et al., 2014; Seinfeld and Pandis, 1998).

The dust occurrence frequency retrieved from spaceborne instruments like the Cloud-Aerosol Lidar with Orthogonal Polarization (CALIOP, Wu et al., 2014) has been previously used to assess the spatial correlation between dust and cloud phase (Choi et al., 2010; Li et al., 2017a; Tan et al., 2014). Two main problems arise from this approach. First, aerosol within

and below thick clouds cannot be detected. Second, low dust concentrations usually fall below the lower detection limit of CALIOP. It has been shown that the CALIOP level 2 data misses about half of the dust aerosol columns detected by the Aerosol Robotic Network (AERONET, Dubovik et al., 2000) when the Aerosol Optical Thickness (AOT) is less than 0.05 (Toth et al., 2018). Additionally, dust loadings simulated by state-of-the-art models show that most of the regions in the southern hemisphere have an annual AOT lower than 0.01 (Ridley et al., 2016).

Ice particles and cloud droplets may coexist in a so-called mixed phase state (Korolev et al., 2017). Mixed-phase clouds with a liquid cloud top and ice virgae beneath are very frequent (Zhang et al., 2010) while cloud tops classified as mixed-phase are much more rare (Huang et al., 2015b; Mülmenstädt et al., 2015). Indeed, supercooled liquid layers at cloud top are generally observed down to temperatures of -25°C (Ansmann et al., 2008; De Boer et al., 2011; Westbrook and Illingworth, 2011). Because ground-based and satellite retrievals are not yet able to accurately estimate cloud phase mass ratios, the Frequency

Phase Ratio (FPR) is often used instead (Cesana et al., 2015; Cesana and Chepfer, 2013; Hu et al., 2010). For satellite retrievals, this is defined as the ratio of ice pixels to total cloudy pixels. Because most cloud phase retrievals are either classified as pure ice or pure supercooled liquid, the average of the FPR represents the ratio of glaciated to total cloud occurrence rather than the actual average cloud phase within. The latter may be misinterpreted as an average ratio of ice to liquid water for typical clouds. Cloud phase in the northern and southern hemispheres has been retrieved in terms of FPR both by ground-based lidar (Kanitz

et al., 2011) and by different spaceborne instruments (Choi et al., 2010; Morrison et al., 2011; Tan et al., 2014; Zhang et al., 2018). These observations showed significant differences between the two hemispheres. It has been suggested in the same studies that such differences are related to the differences in INP concentrations in both hemispheres. Moreover, the local FPR measured at various temperatures between 3°C and -42°C by lidar in Central Europe over a time span of 11 years has been shown to increase with higher dust loadings (Seifert et al., 2010). Spaceborne lidar measurements of cloud phase and aerosol

occurrences show a significant positive spatial correlation between FPR and the frequency of detectable dust, specially at -20°C (Choi et al., 2010; Tan et al., 2014; Zhang et al., 2012, 2015). This spatial correlation has been also found at different atmospheric conditions including humidity, skin temperature, vertical velocity, thermal stability and zonal wind speed (Li et al., 2017a). However, the analysis of the day-to-day variability of cloud phase has received less attention, especially in remote



areas like the Southern Ocean (Vergara-Temprado et al., 2017). Additionally, a more comprehensive and quantitative
assessment of the potential effect of mineral dust on cloud phase is currently lacking. Therefore, in this work the possible effect
of dust aerosol on glaciated stratiform clouds will be assessed based on daily occurrences around the globe between 3°C and
-42°C. For this purpose, the MACC global aerosol reanalysis will be used to sort the cloud phase retrievals of the CALIOP-
GOCCP (Global Climate Model Oriented Cloud Calipso Product, Cesana and Chepfer, 2013) and DARDAR-MASK (raDAR–
liDAR, Delanoë and Hogan, 2008, 2010) satellite products on a day-to-day basis according to the dust mixing-ratio at the
moment of the retrieval, for the period 2007-2010. This method provides a new approach to study the link between dust and
cloud phase variability.

In Sect. 2, the datasets used for the study and the corresponding retrieval algorithms will be presented. In Sect. 3, the
postprocessing of the dataset will be described, including the data structure used, the different filters applied to the data and
the methodology applied to assess the day-to-day correlation between dust and cloud phase. In Sect. 4.1, a short case study
will be presented to compare the different cloud phase products used in the study, then the variability of cloud phase with
respect to latitude and temperature will be briefly assessed in Sect. 4.2 and 4.3. In Sect. 4.4 - 4.6, the main results will be
presented showing the day-to-day correlation between dust and cloud for different latitudes, dust sizes, seasons and temperature
ranges. In Sect. 5, the most important assumptions and sources of uncertainty will be discussed to identify the main limitations
of our approach.

## 2    Data

This section present an overview of the A-Train satellite products and the MACC reanalysis dataset used in the study.

### 2.1    2B-CLDCLASS

Different algorithms exist to classify clouds depending on the properties retrieved by spaceborne instruments (Li et al., 2015).
The CloudSat cloud scenario classification (2B-CLDCLASS, Sassen and Wang, 2008) uses mainly the radar reflectivity
retrieved by the Cloud Profiling Radar (CPR) and the attenuated backscatter signal from CALIOP to classify clouds into 8
different types. These are low level (stratocumulus and stratus), mid-level (altostratus and altocumulus) and high level clouds
(cirrus), and clouds with vertical development including deep convection clouds, cumulus and nimbostratus (Sassen and Wang,
2008). The main criteria for the classification of non-precipitating clouds include the cloud spatial dimension (cloud base, top,
thickness, horizontal extension and homogeneity), radar reflectivity and temperature (from the ECMWF-AUX product).

### 2.2    CALIOP-GOCCP

The CALIOP-GOCCP v.3.0 product (Cesana and Chepfer, 2013) uses the attenuated total backscatter (ATB) and cross-
polarized ATB (ATB⊥) signal from CALIOP to detect cloudy pixels, which are defined as pixels with a scattering ratio



(SR = ATB/ATB$_{mol}$) higher than 5. The cloud volume fraction represents the fraction of cloudy pixels within each gridbox. The product also uses the depolarization ratio of the retrieved signal components to make a decision on cloud-phase (ICE or

LIQUID) based on empirical data. This decision is made for each pixel, with a vertical resolution of 480 m. From this information the FPR is calculated for each 2°x2° gridbox. These gridboxes are then regridded into 3°C bins using the temperature from the Modern Era Retrospective-analysis for Research and Applications (MERRA, Bosilovich et al., 2011) reanalysis.

## 2.3    DARDAR-MASK

The ICARE center's project DARDAR-MASK v1.1.4 (Delanoë and Hogan, 2008, 2010) combines the attenuated backscatter from CALIOP (532 nm ß, sensible to small droplets), the reflectivity from Cloud-profiling Radar (CPR, sensible to larger particles) onboard the CLOUDSAT satellite and the temperature from the ECMWF-AUX product to assess cloud phase. A decision about cloud phase is made at each 60 m vertical pixel, collocated with the CLOUDSAT footprints (approximately 1.1 km horizontal resolution). If the backscatter lidar signal is high (>2·10$^{-5}$ m$^{-1}$ sr$^{-1}$), strongly attenuated (down to at least 10%

in the next 480 m) and penetrates less than 300 m into the cloud, it is assumed that supercooled droplets are present. If this is the case, the pixel is categorized as supercooled or mixed depending on the radar detection, which is assumed a priori to indicate the presence of ice particles. Otherwise the pixel is categorized as ice (Delanoë et al., 2013; Mioche et al., 2014).

## 2.4    MACC

Based on the ECMWF's Integrated Forecasting System (IFS) software the Monitoring Atmospheric Composition and Climate

model (MACC, Eskes et al., 2015) simulate the emission, transport and deposition of various aerosol and trace gases with an output of 1.125° x 1.125° and 60 vertical levels. In this study, we use the model levels of the MACC reanalysis daily product provided by the ECMWF as dataset for dust mixing-ratio and large-scale vertical velocity. Additionally, the pressure levels from the ERA-Interim reanalysis daily product (Dee et al., 2011) will be used in Sect. 5 to provide information about the variability of the relative humidity. The dust emission in the MACC model is parameterized as a function of the 10 m wind

and the vegetation, soil moisture and albedo. The dust loadings are corrected by the assimilation of total AOT at 550 nm from the MODIS instrument on board NASA's Aqua and Terra satellites. Dust deposition is also simulated including dry and wet deposition, in-cloud and below-cloud removal. Finally, dust aerosols are represented through three size bins, with limits at 0.03, 0.55, 0.9 and 20 μm diameter. In this work, we will define as "fine dust" the size bin between 0.03 and 0.55 μm diameter and as "coarse dust" the size bin between 0.9 and 20 μm. Although mostly focused on the northern hemisphere, several studies

have evaluated the simulated dust mixing-ratios from the MACC reanalysis with observations obtaining good results. These observations include ground-based measurements from the AERONET network (Cuevas et al., 2015), shipborne measurements (Ansmann et al., 2017) and satellite retrievals from instruments like CALIOP (Georgoulias et al., 2018).



## 3    Methods

The daily cloud-phase and aerosol products were binned into a 5D-dimensional space of 12 months (e.g. "January" containing

Jan'07, Jan'08, Jan'0 and Jan'10), 10 dust deciles (explained below), 15 temperature bins (of 3 K each), 96 latitudes and 12 longitudes. Within a 12 K range, each latitude band (1.875° width) contains about 1500 to 2000 non-missing datapoints in the mid-latitudes and about 500 to 1500 datapoints in the high-latitudes, with the lowest sample size found for the high southern latitudes (see Fig. 1). To avoid artefacts arising from the averaging of dimensions containing missing values the averaging order was defined (from first to last dimension to be averaged) as *longitude, month, decile, latitude, temperature*. Latitude and

temperature are averaged last given the higher associated correlation with cloud-phase (Sect. 4.2 - 4.3 of this work; Choi et al., 2010; Tan et al., 2014). Further aspects concerning the sample size of the data will be discussed in Sect. 5.

The horizontal grid used correspond to a Gaussian T63 grid (1.875°x1.875°) and temperature bins of 3 K each are used as vertical coordinate. This horizontal resolution is commonly used in Global Climate Models (GCMs, Randall et al., 2007) and

facilitates the future comparisons with global simulations of cloud phase. For the temperature bins, the respective temperature dataset of each product is used (DARDAR and CLDCLASS: ECMWF-AUX, GOCCP: MERRA).

Only night-time values were used to avoid the effects of sunlight scattering in the detection of the CALIOP lidar signal. Also only pixels where the CALIOP retrieval was classified as cloudy (SR > 5) were used to avoid biases in the radar retrievals at lidar fully attenuated pixels. Only non-precipitating clouds were included. Non-precipitating clouds were classified as

gridboxes with less than 10 % precipitating pixels (from the total cloudy pixels), as retrieved by the CLOUDSAT 2B-CLDCLASS product. This was done to avoid the effects of precipitation on dust wet deposition as well as to exclude the interference of rain droplets on the radar reflectivity. The same filters (night-time, lidar-not-fully-attenuated and non-precipitating) was applied to both CALIOP-GOCCP and DARDAR-MASK cloud products.

Huang et al., (2012; 2015) compared the cloud phase retrieved by the DARDAR-MASK and the CALIOP level 2 cloud layer

product (Hu et al., 2009) for low-level clouds in the Southern Ocean (at 40-60°S, 125-145°W and at 40-65°S, 100-160°E). In this study, most clouds are classified as ice or mixed-phase by the DARDAR-MASK at -10°C, while almost all clouds are classified as liquid by CALIOP at equal temperature. This discrepancy is attributed to the low sensitivity of CALIOP to ice particles below liquid cloud tops, to different approaches used in the DARDAR-MASK and CALIOP-GOCCP classification algorithms, and to high clouds interfering with the detection of low clouds.

To allow for a meaningful comparison between the DARDAR-MASK and CALIOP-GOCCP cloud phase products, we defined a new phase ratio from the DARDAR-MASK classification. In this alternative definition, the gridboxes (T63 resolution, 1.875°x1.875°x3 K) with only ice pixels (fully glaciated) are considered as ice clouds and else as supercooled liquid. The purpose is to reclassify mixed-phase clouds, which are mostly only detected as such by the DARDAR-MASK product, into the liquid classification as they are usually classified from the CALIOP-GOCCP product.





In detail, the DARDAR-MASK categories were first dichotomized to a single cloud-phase class (ICE or LIQUID) per pixel, reclassifying MIXED pixels into the LIQUID category. The ratio of ice pixels to the total number of pixels within each gridbox is then defined as $FPR_{DARDAR}$. We then calculate an alternative $FPR\_ALT_{DARDAR}$ with the following rule: The gridbox cloud phase is set to 1 (fully glaciated cloud) if all cloud pixels in the gridbox are classified as ICE and 0 (not fully glaciated cloud) otherwise. This is equivalent to neglecting all mixed-phase gridboxes and setting them to the LIQUID class. We name the

result of this last reclassification $FPR\_ALT_{DARDAR}$. We note that additionally, several clouds almost entirely composed by ice particles (as classified from the DARDAR-MASK), are classified as liquid clouds after this reclassification if they share a gridbox with cloud pixels classified as supercooled liquid. The variables $FPR_{GOCCP}$, $FPR_{DARDAR}$ and $FPR\_ALT_{DARDAR}$ will be compared in detail in Sect. 4.1 and 4.2.

We use the cloud volume fraction (also known as three-dimensional or 3-D cloud fraction, Chepfer et al., 2010, 2013; Li et

al., 2017a; Yin et al., 2015) retrieved by the CALIOP-GOCCP product as a weight in all averages throughout the analysis. This also shifts the focus of the study towards clouds with larger spatial extension (vertical as well as horizontal). The use of the GOCCP cloud volume fraction as weight also introduces a bias towards cloud tops in thick clouds because of the lidar signal attenuation with depth. On the other hand, this weighting allows a more accurate comparison between both cloud phase products by neglecting ice clouds (or virgae) filling only small fractions of a gridbox, which may be missed by CALIOP but

detected by the CPR.

For each volume gridbox (latitude, longitude, temperature) the complete time span 2007-2010 is used to determine the time deciles of dust mixing-ratio using the aerosol data from the MACC reanalysis. These deciles are used to sort the daily data (including FPR) depending on the daily dust mixing-ratio into 10 different decile ranks. These ranks correspond to dust-mixing-ratio bins, from now on simply "deciles". Then, for each 3 K temperature bin, for each month of the year and for each

gridbox the daily data is averaged (weighted) inside each dust decile. This is done as a multiyear average (2007-2010), e.g. January containing Jan'07, Jan'08, Jan'09 and Jan'10. The resulting field contains one extra dimension for each gridbox (month, dust decile, temperature, latitude, longitude).

The relatively high grid resolution (1.875°x 1.875°) result in a large fraction of missing data within the 5-D field. Therefore, the data is aggregated into coarser gridboxes (30°x1.875°) to optimize the number of different satellite swaths contained in

each gridbox. The resulting field contains 5-dimensional variables with 12 months, 10 dust time deciles (calculated for each gridbox), 15 temperature bins (at each 3°C from 3°C to -42°C), 96 latitudes and 12 longitudes covering the globe.

In Sect. 4.1, to differentiate between clear sky and liquid phase the adjusted ice volume fraction

$$FPR^* = (2 \cdot FPR - 1) \cdot cvf \tag{3.1}$$

is used instead of *FPR,* with *cvf* the cloud volume fraction obtained from the GOCCP product. In Sect. 4.2, to assess the

correlation between the mixing dust ratio *m* and the *FPR*, the normalized covariance is used, defined as

$$\overline{COV} = \frac{COVAR(m,FPR)}{\sqrt{VAR(m)}} \tag{3.2}$$





with *VAR(x)* and *COVAR(x,y)* the time variance and covariance, respectively. In Sect 4, all FPR averages (for each dimension) were calculated as

$$FPR_{avg} = \frac{\sum(cvf_i \cdot FPR_i)}{\sum cvf_i} \tag{3.3}$$

with $cvf_i$ the stratiform cloud volume fraction in each gridbox, calculated as

$$cvf_i = cvf_{i,altostratus} + cvf_{i,cirrus} + cvf_{i,altocumulus} + cvf_{i,stratocumulus} \tag{3.4}$$

In Sect 4.5-6, a linear regression was fitted to the results with the form

$$FPR = A \cdot x + FPR_0 \tag{3.5}$$

With $x = m$ and $x = m^{2/3}$. The adjusted correlation coefficient $r^2_a$ of the linear regression was calculated as

$$r_a^2 = 1 - \frac{(1-r^2)(n-1)}{(n-2)} \tag{3.6}$$

with r the Pearson correlation coefficient for one predictor and $n=10$ the number of datapoints.

## 4    Results

### 4.1    Case study

Fig. 2 shows a case study at 9:50 UTC Dec 14, 2010 over the Southern Ocean for temperatures between -42°C and +3°C. The
temperature levels are determined with the ECMWF-AUX product. This A-train segment has been already chosen for a previous case study by Huang et al. (2015) due to the great variety of cloud types it contains. Fig. 2a-b show for the same segment the cloud volume cover (CALIOP-GOCCP) of clouds classified (2B-CLDCLASS) as cirrus or altocumulus (Fig. 2a) and as altostratus or stratocumulus (Fig. 2b). These cloud types are frequently shallow enough to be penetrated by lidar and radar systems and are therefore a good target to study cloud glaciation processes (Bühl et al., 2016; D.Zhang et al., 2010b).
Fig. 2c shows the mixing-ratio of fine (0.03µm-0.55µm) dust aerosol (MACC reanalysis) for the same vertical plane. Fig. 2d-f shows the FPR* (see Sect. 3) which is weighted by cloud volume fraction to highlight the phase of the clouds shown in Fig. 2a-b.

Some major differences can be observed between the three FPR variables in Fig. 2d-f. For the altocumulus cloud at 35-40°S and +3°C to -6°C, the ice virgae falling from the cloud (FPR$_{DARDAR}$) are clearly missed in the FPR$_{GOCCP}$. Because this study
aims at assessing the occurrence of fully glaciated clouds, such mixed-phase clouds are then reclassified in FPR_ALT$_{DARDAR}$ as liquid clouds. A similar case is observed for the stratocumulus clouds at 50-55°S and +3°C to -6°C, and for the altostratus at 35-45°S below -20°C (at higher temperatures). Finally, the cirrus clouds above -33°C remain nearly unaffected by the reclassification in FPR_ALT$_{DARDAR}$ as it is classified as fully glaciated.



## 4.2    Temperature dependence

Mixed-phase clouds between 0°C and -30°C are usually topped by a liquid layer (Ansmann et al., 2008; De Boer et al., 2011; Westbrook and Heymsfield, 2011). Below this layer there is often a thicker layer containing ice particles. Because the CPR is more sensitive to larger particles, this results in a large fraction of the underlying cloud classified as ice in the DARDAR-MASK. At the same time, the CALIOP-GOCCP algorithm usually classifies the whole cloud layer as liquid (Huang et al., 2012; 2015). Below this typical liquid cloud top, the CALIOP backscatter signal is usually already strongly attenuated and

often cannot be used to detect underlying ice layers. This results in a larger FPR retrieved from the DARDAR-MASK compared to the CALIOP-GOCCP product, especially below liquid cloud tops.

Fig. 3 shows that the global average of stratiform $FPR_{GOCCP}$ as a function of temperature from -42°C to 3°C decreases roughly from 100 % at -40.5°C to about 20 % at -1.5°C and down to 0 % at +1.5°C. This temperature dependence between -42°C and 0°C is also observed within a wide range of GCM parameterizations (Cesana et al., 2015), and in ground-based (Kanitz et al.,

2011), spaceborne lidar measurements (Tan et al., 2014) as well as in aircraft measurements (McCoy et al., 2016). However, for the same temperature range the $FPR_{DARDAR}$ only decreases down to 60 % at 1,5°C. The existence of ice at this temperature is partly due to the melting layer being set to a wet-bulb temperature ($T_w$) of 0°C in the DARDAR-MASK algorithm, allowing the categorization of ice water above 0°C dry-bulb temperatures (denominated simply "temperature" in this work). In contrast, $FPR\_ALT_{DARDAR}$ follows very closely the pattern of the $FPR_{GOCCP}$ down to -1,5°C. The differences of the global averaged

$FPR\_ALT_{DARDAR}$ and $FPR_{GOCCP}$ are less than 10 % (absolute difference) for all temperature bins between -42°C and 0°C. This shows that the temperature dependence of the reclassified variable $FPR\_ALT_{DARDAR}$ agrees with the temperature dependence observed for $FPR_{GOCCP}$. Therefore, for the rest of the study, only $FPR\_ALT_{DARDAR}$ and $FPR_{GOCCP}$ will be included.

Additionally, the average fine dust mixing-ratio is also shown Fig. 3. At a temperature of 0°C (note the logarithmic right y-axis) the mixing-ratio is five times higher than at -42°C. This reflects the fact that dust mixing-ratios are higher near the

surface than at high altitudes where temperatures are lower. As a result, across the temperature range of heterogeneous freezing the global FPR appears to be negatively correlated to dust mixing-ratio. Of course, this results alone from the relationship between temperature and dust, and it is therefore a good example of a correlation (between dust and FPR) without a direct causality. In the following of this study, we will explore the correlations between dust aerosol and the occurrence of ice clouds more thoroughly.

## 240    4.3    Zonal dependence

For both temperature ranges shown in Fig. 4 the maximum of FPR is located near the Equator. Similarly, for both temperature ranges the minima are observed towards the high latitudes. Fig. 4a shows the latitudinal dependence of dust and cloud phase at -30°C (averaged from -36°C to -24°C). The $FPR_{GOCCP}$ has two local maxima with values 76 % and 84 % near 39°S and 39°N, respectively. Similar local maxima are observed for the $FPR\_ALT_{DARDAR}$ but at higher latitudes, at 61°S and 61°N with

values 69 % and 74 %. At this temperature both products show a higher FPR in the northern hemisphere than in the southern



hemisphere, in particular for the high latitudes. This higher FPR coincides with the higher average dust mixing-ratio in the northern hemisphere. Such positive spatial correlations between FPR and dust aerosol have been already pointed out using the dust occurrence frequency derived from CALIOP (Choi et al., 2010; Tan et al., 2014; Zhang et al., 2012).

In comparison, the differences between $FPR_{GOCCP}$ and $FPR\_ALT_{DARDAR}$ at -15°C (averaged from -21°C to -9°C) are much

lower than at -30°C as shown in Fig. 4b. However, the $FPR_{GOCCP}$ is lower than the $FPR\_ALT_{DARDAR}$ at the southern mid-latitudes and northern high-latitudes. For both variables, a local minimum near 73°S is followed by a steep increase at 84°S. The error bars for the latter latitude are possibly a result of the low sample size in this region, as mentioned in Sect. 2. However, a higher FPR in the southern than in the northern polar region is consistent with the fraction of ice clouds reported previously in the literature (Huang et al., 2015a; Mülmenstädt et al., 2015)

Some local minima and maxima of the FPR at both temperature ranges are located near the large-scale downdrafts and updrafts, respectively, of the mean-zonal circulation. Indeed, the pattern of the mean large-scale vertical velocity (MACC reanalysis) of the stratiform clouds studied is particularly similar to the FPR at -15°C. Moreover, the spatial correlation between large-scale updraft velocity at 500 hpa has been shown to be positively correlated to the occurrence of ice clouds at -20°C (Li et al., 2017a). In this case, both the dust mixing-ratio and the large-scale vertical velocity seem to be positively correlated to FPR.

However, to understand which (if any) of these variables drives the freezing processes inside the cloud is a much more complex problem of ongoing debate (Sullivan et al., 2016).

## 4.4   Day-to-day variability

Fig. 5 shows the normalized day-to-day covariance between fine dust mixing-ratio and $FPR_{GOCCP}$. This covariance is calculated with the daily values of each specific month of the year. Therefore, it can be understood as a time covariance independent of

the seasonal covariance. This was done to avoid the interferences from the seasonal variability (e.g. peak of dust mixing at northern mid-latitudes ratio in boreal spring) and to focus on the different dust aerosol loadings at similar cloud regimes (e.g. stratiform clouds in winter). The normalized covariance is derived relative to the standard deviation of dust mixing-ratio in each specific gridbox. Therefore, the normalized FPR can be understood as the expected change of FPR given a typical increase of the daily dust mixing-ratio relative to the monthly average.

It can be seen from Fig. 5a that the annual mean of the normalized covariance at -15°C is mostly positive around the globe. The covariance is also considerably higher at the mid and high latitudes and its zonal variability is low compared to its meridional variability.

This meridional pattern can be clearly observed in Fig. 5b, where the variations with temperature are also shown. The covariance in both hemispheres is found to be highest between -10°C and -30°C. More importantly, between -9°C and -21°C

the calculated covariance is larger for the southern hemisphere than for the northern hemisphere. This is clearly in contradiction to the notion that the INP activity of mineral dust is of secondary importance given the low dust aerosol concentrations in the





southern hemisphere (Burrows et al., 2013; Kanitz et al., 2011). Nevertheless, recent studies have acknowledged that the importance of mineral dust in the southern latitudes still cannot be ruled out (Vergara-Temprado et al., 2017).

In Fig. 5c the seasonal variations of the day-to-day covariance are shown, indicating a rather low dependence between the
day-to-day covariance and season. However, a slight increase in the day-to-day covariance can be observed for boreal spring and autumn between 40-80°N. Such differences may be related to differences in the cloud regimes for different seasons. Further investigations are required to clarify the importance and source of such differences.

### 4.5    Influence of dust particle size and mixing-ratio

The results presented in the previous section indicate that the intensity of the day-to-day variations in dust mixing-ratio are
positively correlated with changes in the FPR. It is therefore interesting to assess in detail how the FPR changes for specific dust mixing-ratios.

In Fig. 6, the FPR associated with the calculated deciles of dust mixing-ratio at the mid- and high-latitudes are shown.

As expected from the previous section, Fig. 6a shows that at -15°C the mean $\text{FPR}_{\text{GOCCP}}$ is on average positively correlated with the fine dust mixing-ratio from the MACC reanalysis. Here, each latitude band is represented by ten datapoints, each
corresponding to the average day-to-day deciles of fine dust mixing-ratio. As described in Sect. 3, these deciles are calculated to sort the complete period 2007-2010 into ten ranks for each gridbox and temperature bin. The $\text{FPR}_{\text{GOCCP}}$ appears to be sensitive even to very low mixing-ratios of fine dust. This can be observed in particular at the sub-polar southern latitudes.

For the 60-90°S latitude band, the $\text{FPR}_{\text{GOCCP}}$ at a dust mixing-ratio of 0.1 µg kg$^{-1}$ is 30 %, which is notably higher than the 21 % at 0.01 µg kg$^{-1}$. However, these dust mixing-ratio values in the southern hemisphere are considerably lower than at the
northern mid-latitudes, which are typically in the order of 1 µg kg$^{-1}$. Such low dust mixing-ratios of fine dust in the southern hemisphere often lie below the lower detection limit of CALIOP level 2 data (Toth et al., 2018), and would be classified merely as "clear-sky" conditions. This justifies the use of the MACC reanalysis data for obtaining dust information for this study instead of relying on satellite retrievals (see Sect. 5 for a discussion of the performance of the MACC model).

The $\text{FPR}_{\text{GOCCP}}$ behaves in a similar way for increasing dust mixing-ratios of fine dust in both mid- and high latitudes of both
hemispheres despite the differences for the average fine dust mixing-ratio. To highlight these patterns, a linear regression was fit to all curves using $m$ and $m^{2/3}$ as predictor, with $m$ the dust mixing-ratio. The latter variable, $m^{2/3}$, has been used in the past to model the surface area weight for contact and immersion freezing (Hoose et al., 2008). Table 1 shows the regression and correlation coefficients for $m^{2/3}$ as predictor for both $\text{FPR}_{\text{GOCCP}}$ and $\text{FPR\_ALT}_{\text{DARDAR}}$. The use of $m^{2/3}$ as predictor results in higher correlation coefficients at -15°C for most (13/16) of the regressions compared to $m$ as predictor. Assuming spherical
particles of the same size at constant particle and air densities, the particle surface area concentration can be both expressed in terms of $(m \cdot r^{-1})$, with $r$ the particle radius, or in terms of $(m^{2/3} \cdot \sqrt[3]{n})$ with $n$ the particle number. We therefore speculate that the better regressions fit for $m^{2/3}$ as predictor may be associated to variations in dust particle size controlling the surface area concentration for each size bin. The higher surface area concentration (Atkinson et al., 2013; Niedermeier et al., 2015)





could then explain the higher occurrence of ice clouds. We note, however, that more evidence is needed in order to support

such hypothesis. The regression coefficients shown in Table 1 for fine dust can be useful to interpret the different sensibilities to dust mixing-ratio observed at different latitudes in Fig. 6a. For instance, the increase of FPR$_{\text{GOCCP}}$ at low mixing-ratios (0.01 - 0.1 µg kg$^{-1}$) of fine dust at 60-90°S compared to a similar increase at 30-60°N with higher mixing-ratios (0.1 - 1 µg kg$^{-1}$) is reflected by a regression coefficient ten times larger for 60-90°S (0.386) than for 30-60°N (0.042). In comparison, the coefficients for 30-60°S (0.100) and 60-90°N (0.102) reflect the similar mixing-ratio for which the FPR$_{\text{GOCCP}}$ increase is

observed for these latitudes.

Among the four latitude bands shown in Fig. 6a, the fine dust mixing-ratio ranges over three orders of magnitude. In the same range of dust loadings, the FPR increases from 19 % at the lowest decile of the southern high-latitudes to 40 % in the highest decile of the northern high-latitudes. A similar increase of cloud FPR$_{\text{GOCCP}}$ at -15°C from 45 to 65 % was reported for summarized 11 years ground-based lidar measurements in Leipzig, Germany (Seifert et al., 2010) for simulated dust

concentrations (note the different units) of 0.001 µg m$^{-3}$ to 2 µg m$^{-3}$. This further suggests a common mechanism controlling the correlation between FPR and dust at different latitudes.

Several experimental studies have studied the size-dependence of dust freezing efficiency (Hartmann et al., 2016; Lüönd et al., 2010; Welti et al., 2009). These studies show that the ice fraction for single dust particles immersed in supercooled droplets scales with particle size. However, the comparison of these three studies in Hartmann et al. (2016) shows no considerable

differences in the ice nucleation surface site density for particles between 300 nm and 1000 nm. Furthermore, the relative humidity with respect to ice (RH$_{ice}$) required for the INP activation of different compositions and sizes of dust have been also studied experimentally (Welti et al., 2009). In such experiments, the required RH$_{ice}$ has been shown to be lower for larger particle size at temperatures of -30°C and below, but similar for temperatures of -25°C and -20°C.

Fig. 6b shows the mean FPR$_{\text{GOCCP}}$ for the day-to-day deciles of coarse dust of the MACC reanalysis. Similar to the fine dust

deciles, the FPR$_{\text{GOCCP}}$ increases in the mid- and high-latitudes for higher coarse dust mixing-ratios in both hemispheres. However, the differences of FPR$_{\text{GOCCP}}$ between low and high coarse dust deciles are considerably lower than the ones for fine dust. A suitable measure of this difference is the normalized day-to-day covariance $\overline{COV}$ for both fine and coarse dust mixing-ratio, as mentioned in Sect. 4.4. The ratio between $\overline{COV}_{fine,GOCCP}$ and $\overline{COV}_{coarse,GOCCP}$ is shown in Table 1 and can be understood as the ratio between the FPR variability explained by fine dust relative to coarse dust. This ratio is larger than one

($\overline{COV}_{fine,GOCCP} > \overline{COV}_{coarse,GOCCP}$) for all four latitude bands studied. The largest ratio is observed for 30-60°S (2.08) and the lowest for 30-60°N (1.10). This is an unexpected result, given the number of studies that have associated cloud glaciation to coarse dust, in particular at the northern mid-latitudes (Ansmann et al., 2008; DeMott et al., 2010; Li et al., 2017b; Richardson et al., 2007; Stith et al., 2009).



### 4.6    Influence of temperature and season on the day-to-day variability of FPR

In the previous section, the relationship between the day-to-day dust deciles (MACC) and the FPR$_{GOCCP}$ was assessed in detail for -15°C. In the following section, the same analysis will be repeated to assess the dependence between FPR$_{GOCCP}$ and fine dust for other subsets of interest motivated by the covariance analysis in Sect. 4.4. In particular, the FPR$_{GOCCP}$ at -30°C is presented in Fig. 7a for the same latitude bands assessed in Sect. 4.5.

In contrast to the FPR$_{GOCCP}$ at -15°C, the correlation coefficients for the regressions fitted to the FPR$_{GOCCP}$ and fine dust mixing-
ratio at -30°C (Table 1) were only significant ($r_a^2 > 0.8$) for 30-60°N. For this latitude band, the normalized covariance for coarse dust was higher than for fine dust. This agrees with the results of Welti et al. (2009), who showed that the required $RH_{ice}$ for the INP activation of dust is lower for larger particles at -30°C. Despite the low correlation coefficients, for all latitude bands a relative increase in FPR$_{GOCCP}$ can be observed for the last fine dust deciles at each latitude band.

One final aspect to consider is the seasonal dependence of the day-to-day variability of the FPR. This is particularly
challenging, due to the changes in the mean dust mixing-ratio between seasons. Because the dust deciles (as explained in Sect .3) are calculated using the whole period 2007-2010, this means that the sample size within each dust decile changes slightly from season to season. For instance, this issue is reflected in the standard deviation of FPR$_{GOCCP}$ at the lowest fine dust deciles in each 40-80°N/S latitude band (error bars in Fig. 7b). Here, the lower number of days with relative low fine dust mixing-ratio during summer of both hemispheres results in a larger standard deviation of the FPR$_{GOCCP}$ in the first decile.
Despite these difficulties, it can be seen from Fig. 7b that the differences between the FPR$_{GOCCP}$ at -15°C during summer and winter are low compared to the differences between fine dust deciles for both hemispheres.

Although the INP activity of mineral dust may offer a promising explanation, some other effects may also explain the results presented until this point. Therefore, the next section will focus on discussing the limitations of our approach. Afterwards, the possible interferences from processes other than the INP activity of mineral dust will be discussed.

## 5    Assumptions and uncertainties

In the analysis presented above, certain assumptions were made to assess the potential effect of fine dust on cloud phase. In this section, these assumptions and the uncertainties that arise from them and the subsequent limitations of the resulting interpretation will be discussed.

Despite the long period (2007-2010) used in the study, an important fraction of the dataset analysed contains missing-values
across the 5D-space (10 dust deciles, 12 months, 15 temperature bins, 96 latitudes and 12 longitudes). For instance, the reduced sample size for the high-latitudes (as shown in Sect. 2) introduces an important bias towards the respective winter-time because very few night-time retrievals are available in the respective summer-time. The results suggest that this bias should not have a big impact on the resulting day-to-day variability analysis. This is because the seasonal variability observed in Sect. 4.6 was not found to be a dominating factor in the day-to-day impact of dust mixing-ratio on FPR.





A high dust mixing-ratio simulated in a gridbox indicated as cloudy by satellite observations does not ensure that the dust particles are actually mixed in the cloud. The spatial subgrid-distribution of dust relative to the cloud position (subgrid) remains unresolved. Higher dust mixing-ratios should be interpreted as a higher probability that a significant amount of dust was mixed with a certain cloud. This may have happened during or before being observed by the satellite instrumentation. The latter assuming that both cloud and dust followed the same trajectory. This results in a high level of uncertainty to determine the

coincidence of dust and clouds for a combination of modelled dust and satellite-retrieved cloud information (Li et al., 2017b). It has been shown, that global models, including the MACC model, underestimate the coarse dust fraction in relation to the fine dust fraction (Ansmann et al., 2017; Kok, 2011). This underestimation would affect the regression coefficients presented in Sect. 4.4. However, the authors expect the relative changes in fine and coarse mixing-ratio to be affected to a lesser extent by such overestimation. It is expected that the next generation of atmospheric composition reanalysis (Flemming et al., 2017)

provide a better estimation of dust aerosol size distribution.

Furthermore, it cannot be ruled out that the increase in ice cloud occurrence in the southern hemisphere for higher dust mixing-ratios arise from another type of aerosol like biogenic (Burrows et al., 2013; O'Sullivan et al., 2018; Petters and Wright, 2015) or background free-tropospheric (Lacher et al., 2018) aerosol being misclassified as mineral dust. Similarly, the possible correlation between ice cloud occurrence and the emission and transport of dust aerosol should be also further investigated

(e.g. dusty air masses from land are usually warmer and drier). Another interesting explanation of the results presented in this study includes the possible mixture between mineral dust particles with ice nucleation active macromolecules (Augustin-Bauditz et al., 2016). Such particles have been found to be of the size of few 10 nm (Fröhlich-Nowoisky et al., 2015). This could therefore explain the unexpected findings with respect to the importance of fine dust over coarse dust aerosol for cloud glaciation, as presented in this study.

Relative humidity (RH) is, next to the temperature, one of the main factors in the initiation of ice nucleation in laboratory studies (Hoose and Möhler, 2012; Welti et al., 2009). The correlation between fine dust mixing-ratio from the MACC reanalysis and the RH from the ERA Interim reanalysis (weighted by cloud volume fraction) was found to be significant and negative (Fig. 8a). We expect, however, that the impact of the decreasing RH would lead to an underestimation of the role of dust mixing-ratio in cloud glaciation. We note that the RH from the ERA Interim reanalysis represents the conditions at a

large-scale and not the conditions at a specific location and the moment of the interaction between dust aerosol and supercooled cloud droplets. Because of this issue, the role of subgrid RH variability in the results cannot be assessed. It remains also unclear whether the relation between RH and dust is connected to the observed correlation between dust and the isotherm height (Fig. 8b).

Indeed, the significant positive correlation between dust aerosol mixing-ratio and the height of the isotherms (weighted by

cloud volume fraction) is also an important source of uncertainty. Shifts in the isotherm height, due to warm dry continental air masses carrying high dust loadings or the heating effect of absorbing dust aerosol, may indeed cause a dynamic response from the atmosphere leading to higher isotherms (Gómez-Amo et al., 2014; Meloni et al., 2015; Peris-Ferrús et al., 2017). This



could lead certain clouds to be detected in a higher temperature bin after being glaciated at lower temperatures, thus erroneously suggesting an enhanced glaciation occurrence at higher temperatures. It is crucial for future studies to acknowledge this
challenge when studying the occurrence of ice clouds at a certain isotherm.

Perhaps the main difficulty of attributing the observed changes to dust concentration variability is the positive correlation between dust and the vertical velocity. Fig. 8c shows the large-scale vertical velocity from the MACC reanalysis against the fine dust deciles at -15°C. The generation of supersaturation is generally associated with stronger updrafts. Supersaturation over water is a necessary condition for contact and especially immersion freezing mode, which is believed to be the most
important freezing mode in cloud glaciation, specially at temperatures above -25°C (Westbrook and Illingworth, 2011). Li et al. (2017a), showed using the monthly values of the CALIOP-GOCCP product that the FPR at -10, -20 and -30°C is positively (spatial) correlated with the large-scale vertical velocity (at 500 hpa) from the Era-Interim reanalysis product. The same study showed using the CALIOP level 2 aerosol layer product that this correlation is similar for different relative aerosol frequencies (dust, polluted dust and smoke combined). However, in the same study the increase in FPR for higher vertical velocities (higher
updrafts) was found to be higher for lower relative aerosol frequencies. These lower relative aerosol frequencies correspond to the remote regions in the high-latitudes, for which the largest sensitivities to dust mixing-ratio were found in our study. Nevertheless, it should also be considered, that regions of large-scale lifting are in general favouring the formation of horizontally and vertically widespread cirrus. This could lead to a bias which would mask the relation between dust and FPR. We note that in our study not enough evidence was found to discard an interference between dust and atmospheric dynamics.
Such an interference could partly result from the observed correlation between cloud phase and fine dust mixing-ratio being attributable to processes other than the INP activity of mineral dust. To understand whether INP (microphysics) or dynamics dominates the ice nucleation process is actually a topic of ongoing research (Sullivan et al., 2016). However, the variables found to be correlated to the dust mixing-ratio (RH, vertical velocity and isotherm height) could not be directly linked to the observed increments in FPR. This suggests that even in the scenario where a high influence of atmospheric dynamics in the
results, the effect of dust may still explain an important part of the day-to-day FPR variability.

## 6    Conclusions

For the first time, a reanalysis of atmospheric aerosol composition was combined with two different satellite-retrieved cloud phase products to investigate the potential effect of mineral dust as INP on a global scale and on a day-to-day basis using collocated satellite retrievals of cloud phase for a four-year period (2007-2010). In this study, we studied stratiform clouds
observed at night-time. Our main findings can be summarized as follows:

1. Between -21°C and -9°C, day-to-day increases in fine dust mixing-ratio (from lowermost to highermost decile) were found to be associated with increases in the day-to-day stratiform FPR of +10 % to +18 % in the mid- and high-



latitudes. These increments were found to be almost twice as high when comparing fine dust as when comparing coarse dust mixing-ratios on a day-to-day basis.

2.   The trends of cloud phase associated with relative day-to-day increments in fine dust loading were found to be similar between the mid- and high- latitudes and between southern and northern hemispheres. The trends were best fitted by a linear regression of the form $FPR=A \cdot m^{2/3}+FPR_0$. Moreover, the increments in FPR (from first to last dust decile) were larger in remote regions like the northern and southern high-latitudes, where dust aerosol is believed to play a minor role in cloud glaciation. The observed trends also suggest the existence of different sensitivities to fine dust for

440        different latitude bands. The largest sensitivities were observed for the high-latitudes in the southern hemisphere.

      3.   The interference from atmospheric dynamics related to higher mixing-ratios of dust aerosol could not be entirely discarded. Specifically, the day-to-day variability of the large-scale vertical velocity and in particular of the isotherm height were found to be also correlated with the dust mixing-ratio. Future studies should therefore acknowledge such correlations when assessing the day-to-day variability of cloud phase.

The authors hope that the results of this work will motivate further research. This includes field campaigns in remote regions to study the day-to-day variability of cloud thermodynamic phase and the role of mineral dust in ice formation, satellite-based studies of the resulting changes in radiative fluxes, and modelling studies to test the representation and relevance of specific processes involved in ice formation and mineral dust transport. Such studies will help to improve our understanding of the influence of mineral dust on cloud glaciation and the climate system.

**Author contribution**

DV, IT, BH and PS contributed to the design of the study. DV processed the datasets, performed the analysis, designed the figures and drafted the manuscript. All authors discussed the results and contributed to the final manuscript.

**Competing interests**

The authors declare that they have no conflict of interest.

**Acknowledgments**

We thank the GOCCP project for providing access to the CALIOP-GOCCP gridded cloud phase profiles. We thank the NASA CloudSat project and the CloudSat Data Center for providing access to the 2B-CLDCLASS product. We thank the ICARE Data and Services Center for providing access to the DARDAR and CloudSat data. We also thank the MACC project for providing access to the MACC reanalysis dataset. All datasets used in the analysis are freely available at

http://climserv.ipsl.polytechnique.fr/cfmip-obs/Calipso_goccp_new.html,    http://www.icare.univ-lille1.fr/archive    and





http://apps.ecmwf.int/datasets/data/macc-reanalysis/levtype=ml (last access: 18 October 2018). We thank Dr. Albert Ansmann and Dr. Johannes Mülmenstädt for helpful discussion.

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



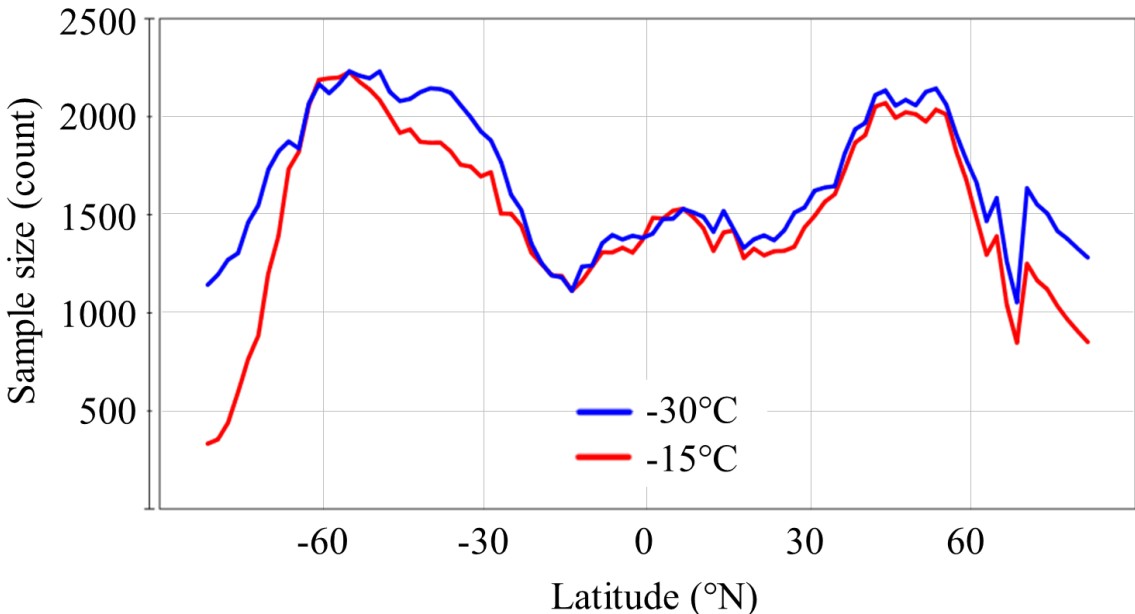

**Figure 1. Sample size of cloud phase (CALIOP-GOCCP) of each latitude band for -15 °C (range -21 °C to -9 °C) and -30 °C (range -36 °C to -24 °C) for the period 2007-2010. Each count corresponds to a 1.875°x30° gridbox in a 3 K temperature bin at a specific month of the year and inside a specific dust decile (calculated relative to the mixing-ratio in each gridbox and temperature bin during 2007-2010). The theoretical maximal sample size for each latitude band is 5760 (12x12x10x4) for a 12 K temperature range.**



**Figure 2. Case study 9:50 UTC Dec 14, 2010. a, b) Cloud volume fraction (GOCCP) for different stratiform clouds (CLOUDSAT cloud classification). c) Fine dust (0.03μm-0.55μm) aerosol mixing-ratio (MACC reanalysis), note the logarithmic scale. (d-f) Adjusted ice occurrence derived from the DARDAR (d, e) and GOCCP (f) products. FPR: Frequency phase ratio (ice pixels/total pixels). In FPR_ALT mixed-phase gridboxes (containing both liquid and ice pixels) are reclassified as ice. White colours represent clear sky. The fields were collocated in a 1.875°x1.875° grid with temperature bins of 3 K each.**




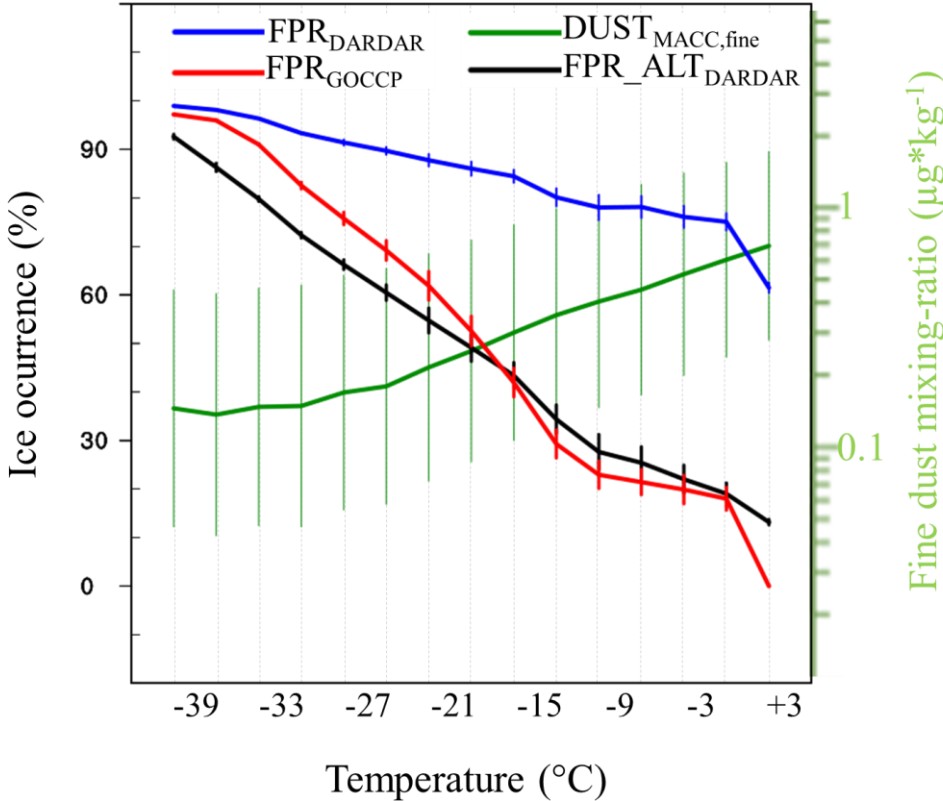

**Figure 3. Global ice cloud occurrence for stratiform clouds (2007-2010). The average FPR is weighted by the volume cloud fraction.
The fine dust mixing-ratio from the MACC reanalysis corresponds to the range 0.03-0.55 µm and is presented on a logarithmic scale
on the right vertical axis. Each temperature bin spans 3 K. The error bars show the mean day-to-day standard deviation between
different fine dust deciles.**

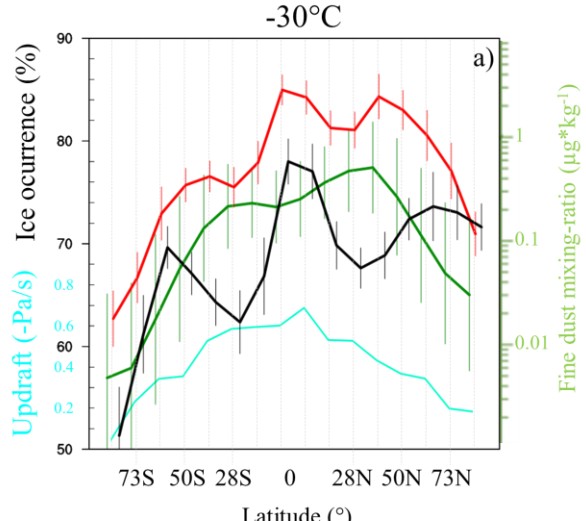
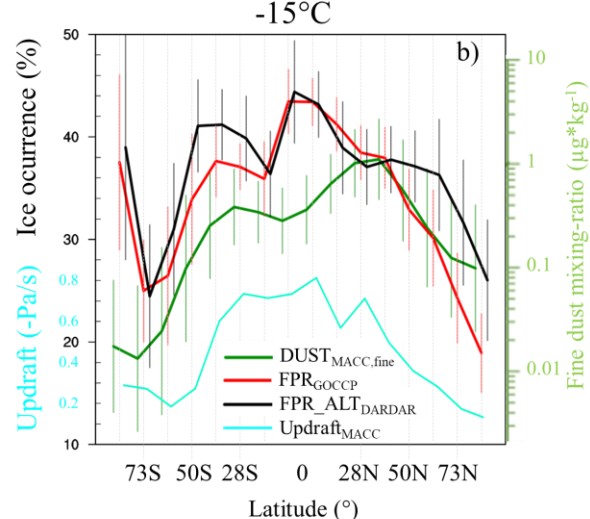

**Figure 4. Zonal mean of stratiform cloud ice occurrence for (a) -30 °C (range -36°C to -24°C) and (b) -15 °C (range -21 °C to -9 °C) averaged over the period 2007-2010. Each datapoint corresponds to a zonal band of 11.25° width. The average fine dust mixing-ratio of each band is also shown on the right vertical axis (note the logarithmic scale). The average large-scale vertical velocity from the MACC reanalysis is also shown (cyan axis on the left of each plot). The error bars show the mean day-to-day standard deviation between different fine dust deciles. The curves for dust and FPR_ALT$_{DARDAR}$ are slightly shifted left and right, respectively, to fit all error bars.**



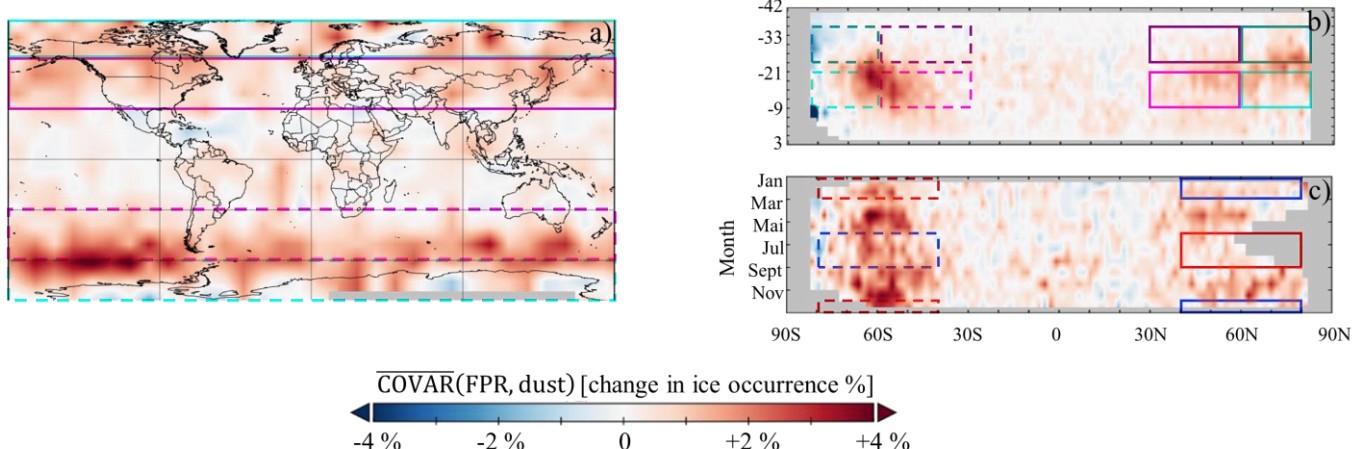

**Figure 5. Day-to-day covariance of fine dust mixing-ratio (MACC) and FPR$_{GOCCP}$ 2007-2010 . The covariance is normalized by the fine dust standard deviation. a) Time mean covariance between 12 monthly covariance values at -15°C (averaged in a 12 K range). b) Time-zonal mean covariance. c) Monthly-zonal mean covariance at -15°C. Red colours denote a high correlation between day-to-day fine dust and ice occurrence. The colour and pattern of the boxes corresponds to the regression lines in Fig. 6 and 7.**



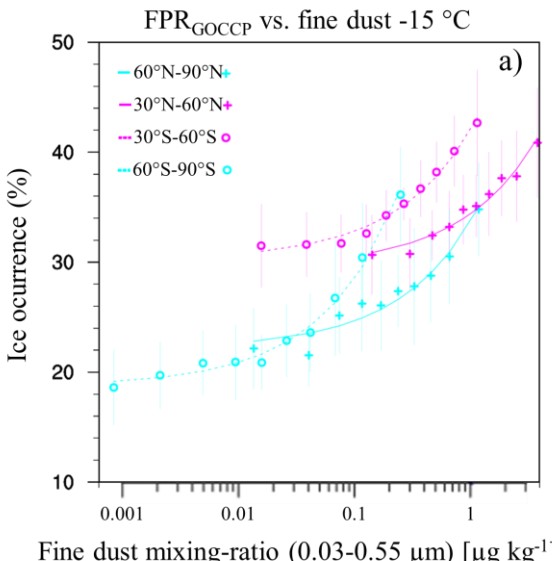

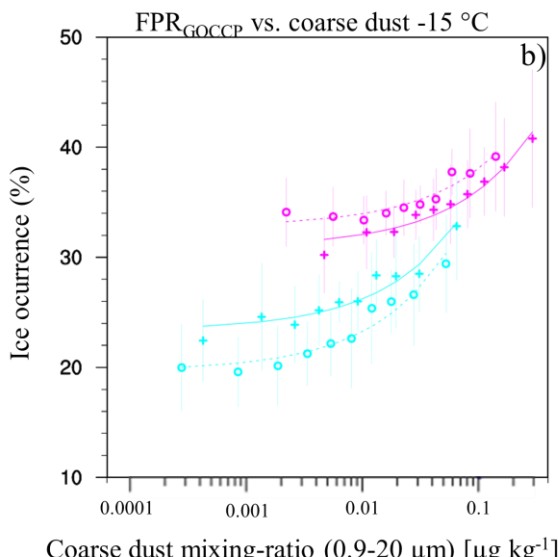

**Figure 6. Average cloud phase for the mid-latitude and high-latitude bands for -15 °C (range -21 °C to -9 °C) in the period 2007-2010. The horizontal axis corresponds to the different time deciles (day-to-day variability) of fine (a) and coarse (b) dust mixing-ratio (MACC), calculated for each each 3°C temperature bin and gridbox (1.875°x1.875°) and averaged along each 12 °C temperature range and latitude band (30°x360°). The error bars are positioned at each dust decile and show the mean zonal standard deviation within each latitude band. The lines represent the linear regressions ($FPR = A \cdot m^{2/3} + FPR_0$ ), see Table 1 for details.**





$$FPR = A\, m^{2/3} + FPR_0 \mid m \text{ in } [\mu g/kg]$$

| Temperature: -15 °C | | Fine (0.03-0.55 µm) | | | Coarse (0.9-20 µm) | | | $\dfrac{\overline{COV}_{fine,GOCCP}}{\overline{COV}_{coarse,GOCCP}}$ |
|---|---|---|---|---|---|---|---|---|
| Variable | band | $A$ | $FPR_0$ | $r_a^2$ | $A$ | $FPR_0$ | $r_a^2$ | |
| FPR_ALT$_{DARDAR}$ | 60-90S | 0.391 | 0.21 | **0.93** | 0.635 | 0.21 | **0.87** | |
| FPR$_{GOCCP}$ | 60-90S | 0.386 | 0.19 | **0.99** | 0.649 | 0.20 | **0.93** | 1.44 |
| | 60-90S | 0.389 | | | 0.642 | | | |
| FPR_ALT$_{DARDAR}$ | 60-90N | 0.130 | 0.27 | **0.98** | 0.725 | 0.28 | **0.94** | |
| FPR$_{GOCCP}$ | 60-90N | 0.100 | 0.22 | **0.93** | 0.514 | 0.23 | **0.91** | 1.25 |
| | 60-90N | 0.115 | | | 0.619 | | | |
| FPR_ALT$_{DARDAR}$ | 30-60S | 0.081 | 0.37 | 0.92 | 0.252 | 0.37 | 0.92 | |
| FPR$_{GOCCP}$ | 30-60S | 0.102 | 0.30 | **0.99** | 0.207 | 0.33 | 0.89 | 2.08 |
| | 30-60S | 0.091 | | | 0.230 | | | |
| FPR_ALT$_{DARDAR}$ | 30-60N | 0.032 | 0.33 | **0.91** | 0.141 | 0.34 | **0.93** | |
| FPR$_{GOCCP}$ | 30-60N | 0.042 | 0.30 | **0.96** | 0.205 | 0.31 | **0.95** | 1.10 |
| | 30-60N | 0.037 | | | 0.173 | | | |
| Temperature: -30 °C | | Fine (0.03-0.55 µm) | | | Coarse (0.9-20 µm) | | | |
| FPR_ALT$_{DARDAR}$ | 30-60N | | 0.70 | 0.00 | 0.189 | 0.81 | **0.95** | |
| FPR$_{GOCCP}$ | 30-60N | 0.034 | 0.81 | 0.89 | 0.106 | 0.69 | 0.59 | 0.88 |
| | 30-60N | 0.034 | | | 0.148 | | | |

**Table 1. Summary of the linear regression results for cloud ice occurrence as a function of dust mixing-ratio.** $A$ is the regression coefficient, $FPR_0$ is the y intercept and $r_a^2$ is the adjusted square of the Pearson correlation coefficient. Bolded values of $r_a^2$ represent the regressions for which $r_a^2$ is higher for $x=m^{2/3}$ than for $x=m$, with $m$ the dust mixing-ratio. For -30° C, only two regressions where significant ($r_a^2 > 0.8$). Underlined values represent the mean between the regression coefficients for each latitude band. The darker green colours correspond to the higher correlation coefficient and the darker blue colours correspond to the higher y intercepts ($FPR_0$).





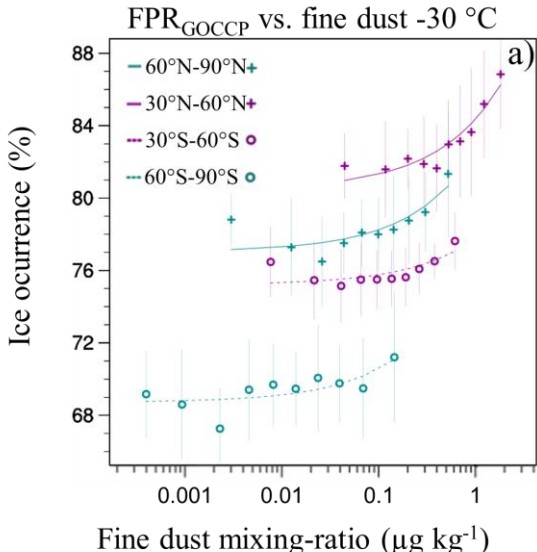
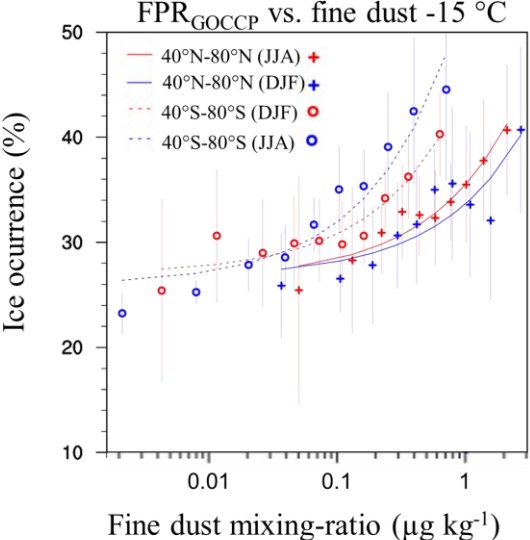

**Figure 7. Same as Fig 6.a but for a) -30 °C (range -36 °C to -24 °C) and b) winter (blue) and summer (red) seasons for the 40-80°N/S latitude bands.**



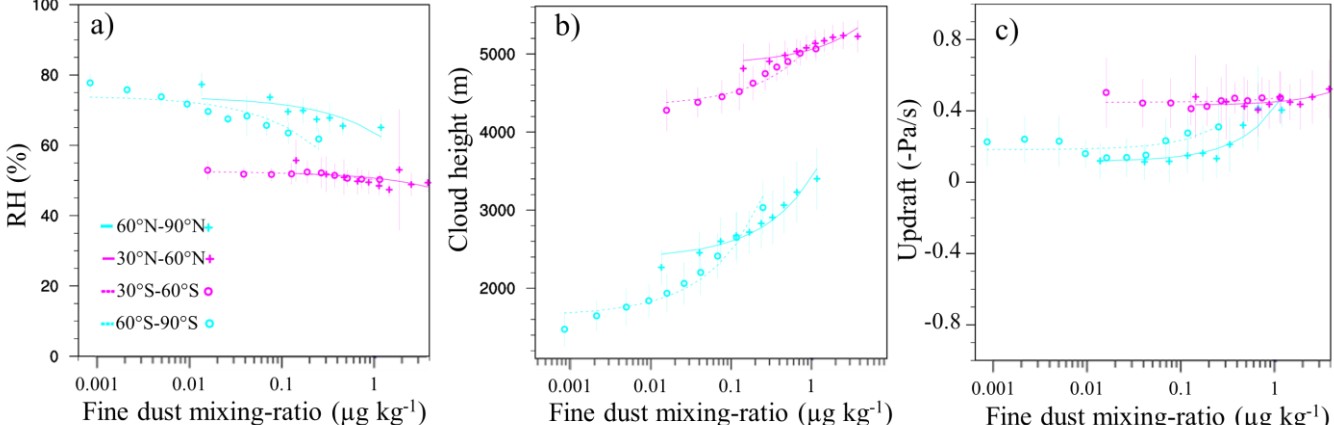

**Figure 8. Same as Fig. 6a but for a) ERA Interim relative humidity, b) ECMWF-AUX isotherm height and c) MACC large-scale vertical velocity at -15°C. The average of each variable is weighted by cloud volume fraction analogous to Fig. 6 and Fig. 7.**