# Peer review of "The impact of mineral dust on the day-to-day variability of stratiform cloud glaciation occurrence"

_Atmospheric Chemistry and Physics, 2018_

## Referee Comment (RC1) · Anonymous Referee #1 · 9 Dec 2018

**Review of:** The impact of mineral dust on the day-to-day variability of stratiform cloud glaciation occurrence

**Authors:** Diego Villanueva, Bernd Heinold, Patric Seifert, Hartwig Deneke, Martin Radenz, Ina Tegen

**General Comments:** This paper explores important relationships between aerosol properties (specifically mineral dust loading), cloud phase, and other dynamic and thermodynamic variables. I commend the authors for attempting to provide a global perspective on this challenging to measure and understand phenomenon. The satellite-based perspective on aerosol cloud interactions can provide more insight into the global extent of aerosol influences than obtainable from measurements from a single observatory. In general, I think that the analysis completed is interesting and potentially sheds some light onto the relationships between mineral dust and cloud properties. Having said that, I do have some reservations and questions about the manuscript, and therefore would like to see some additional work completed before this paper can be accepted for publication. Additionally, I found the manuscript to be very dense, and found myself having to reread sections on a regular basis. I'm not sure whether this was the result of the frequent use of abbreviations, or the writing style, or something else. However it was a challenge to read, which is unfortunate given the science that is in the paper.

**Specific Comments:**
- I wonder to what extent the different products used in this evaluation are consistent with one another. This is particularly a large question for the reanalysis derived estimates of temperature, humidity and vertical velocity. Since the reanalyses rely on models to provide input on clouds and dynamics, the values represented in these products must be internally consistent — however, this may not always match the real atmosphere. This is particularly true in areas of cloud cover, where the model clouds and real clouds may not match in time, location and phase. Therefore, I wonder whether there may be instances where the model (reanalysis)-produced thermodynamic state is inconsistent with the clouds detected from satellite measurements, potentially biasing the evaluation of observed cloud phase into different temperature regimes from reality. Some discussion on the potential for this to occur would be helpful.
- There is a substantial question related to the ability to truly detect relationships between dust and cloud phase in the absence of sufficient constraints on environmental state and dynamics. The co-variability that is demonstrated is interesting, but how can one be sure that this is the result of the aerosol, and not of the dynamical forcing on the cloud? Particularly at high latitudes, where figure 8 appears to show a relationship between dust mixing ratios and vertical velocities, it could be challenging to discern the impacts of the dynamics from those of the aerosols. I will say that it seems to me that the relationship is such that you would not necessarily expect that the dynamics and aerosols would work in the same direction – with increasing mineral dust loading you have increasing vertical velocities (upward), which would support enhanced supersaturation. Therefore,

the cloud liquid would increase in response to the updraft but decrease in response to the mineral dust loading. Assuming I have that correct, this could help to support the idea that the changes are the result of aerosols and not dynamics, but I think that a good amount of discussion on this topic is warranted.

- *Line 129*: What, if any, sensitivity is there to the order of the averaging?
- *Lines 139-140*: Please describe what cloud depth constitutes precipitating vs. non-precipitating in 2B-CLDCLASS
- *Line 179*: 30x1.825 deg — depending on latitude, this can be a very large amount of area. Therefore, I would be concerned about sub-grid variability, particularly at lower latitudes. For example, dust and cloud could be on different sides of a front within a given grid box. This is discussed a bit starting on line 370. Additionally, there could be significant gradients in phase over such a large horizontal extent, particularly around mid-latitude frontal zones, coastlines, etc. Such gradients may have little to do with the dust, but everything to do with changes in forcing.
- Figures 3 and 4: Recommend removing the dotted vertical lines to improve clarity.
- Figure 3: "Error bars" — this isn't really error, is it? Just the standard deviation and variability?
- Figure 4: Same comment about "error bars"

---

## Referee Comment (RC2) · Anonymous Referee #2 · 10 Dec 2018

General comments

The authors analyzed the impact of fine and coarse dust aerosols on the day-to-day variability of stratiform cloud glaciation occurrence by using 4 years' satellite cloud phase products and MACC aerosol reanalysis. Compared to the previous study, this study mainly focused on the day-to-day variability of cloud phase. The results showed that the phases of stratiform clouds is highly dependent on temperature and latitude, and dust aerosol mixing ratio is anti-correlated with the average occurrence of fully glaciated stratiform clouds. Generally speaking, the paper is interesting, and tables and graphics are well constructed. As a result, I am recommending the paper be

accepted with major revisions if the authors response properly my comments. The some main questions and comments I have are listed below in the specific comments to the authors.

Specific Comments:

(1) One of my concerns is: The applicability of MACC aerosol reanalysis, especially over the southern ocean, where ground-based measurements are sparse. There is no direct observation evidence of dust aerosol to prove the applicability of MACC aerosol reanalysis. In addition, the variable used in this study is dust mixing-ratio? Isn't mass concentration? I think that the dust mass concertation should be more proper for this analysis.

(2) One suggestion: The section 3 includes too much information, and it is easy to confuse the readers. May you provide us one table or flow chart? The authors also reconstruct this section to make the method more clear. For example, move the lines 177-195 to the first paragraph, and following sentences interpret these variables.

(3) Line 125: What is the mean of the Jan'0?

(4) Line 140: Please provide the detailed information about the classification of non-precipitation.

(5) For the equations 3.2-3.5, FPR or FPR*?

(6) The main concern is: the authors how to peel off the impact of meteorological condition from dust loading. Because the aerosol and dynamical factor usually are co-variability. Thus, some discussions about dynamical factor are necessary.
* * *

---

## Referee Comment (RC3) · Anonymous Referee #3 · 12 Dec 2018

General comments:

The manuscript presents an interesting study of mineral dust impacts on stratiform cloud glaciation occurrence using satellite remote sensing measurements and MACC aerosol reanalysis daily product. Mineral dust is one of the major atmospheric aerosols and are also efficient INPs. However, it is still challenging to quantify their impacts on ice nucleation in clouds and cloud thermodynamic phase partitioning. Their findings in this study confirm that mineral dust has positive correlations with ice cloud occurrence on a global scale even after limiting other impact factors. The approach presented in this study is very unique. Personally, I like the approach and also am very happy with

their findings. However, many of their discussions are hard to explain the observed features (sometimes contradict with previous studies). Besides, the manuscript is not well organized and poorly written (especially the comparisons between FPRDARDAR and FPRGOCCP) and therefore is hard to read. Overall, I believe serious revisions are needed before the manuscript be accepted by ACP.

Major comments:

Fig1.: 1) About the sample size, what causes the missing datapoints? 2) Why do the high-latitudes have less datapoints than the mid-latitudes? 3) Is the sample size same or similar for DARDAR and MACC?

Fig 2. and related discussion: 1) According to line 143-144 and line 153, "lidar-not-fully-attenuated" and "only ice pixels are considered as ice clouds", I would expect basically DARDAR and GOCCP have very similar FPR. For example, in Fig 2., DARDAR de-tected the ice virga below the liquid layer top. However, generally, the liquid layer at the top will fully attenuate the lidar signal (also as stated in line 119). Therefore, given the filters listed in line 143-144, I do not understand why DARDAR still show an ice layer below; 2) Comparing 2a, b, d, e, and f there are not clouds at latitudes between -40 and -45 and temperatures between -6 and 3, there shows no cloud in 2a, b, and e, but does have FPRDARDAR and FPRGOCCP in 2d and f, what's the reason for the differ-ences? 3) Comparing 2d and e, at latitudes between -40 and -45, the whole profiles shows ice cloud in 2d but it is liquid in figure 2e. Even with all the rules described in line 157-158, it is still difficult to understand why it is liquid in figure 2e (the grid box is still 1.875 x 1.875, right?)

Fig.3: 1) I really object to put dust mixing ratio in this figure. There is no physical rela-tionship between dust mixing-ratio and temperature. Dust mixing ratio decreases with height near the source region and temperature generally decrease with height in the atmosphere. But physically, there is no relationship between temperature and dust! 2) In line 235, "mixing-ratios are higher near the surface than at high altitudes", this is not

true for transported dust layer such as over the high latitude regions. Overall, this study is trying to look at the relationship between pure ice cloud fractions and dust mixing ratios. Therefore, comparing FPRDARDAR and FPRGOCCP has little contribution to the main research topic in this study and the descriptions of converting different FPRs are quite confusing and distracting. In fact, in the proceeding sections, only FPRGOCCP is used most of the time. Therefore, I would suggest just use FPRGOCCP in the study.

Fig. 4: 1) It is surprising that 'maximum of FPR is located near the Equator'. Kanitz et al. (2011) showed that Northern mid-latitudes have much higher ice-containing cloud fraction than the tropical regions. Do the authors have speculations why the results are so different? Line 251, what's the reason for the steep increase of FPR at 84o? 2) Does this steep increase of FPR only occur at -15 C? How about at -20, -25 C? 3) Line 254, do the referred studies also comparing the fraction of ice clouds at the same temperature? It is really hard to explain why southern polar regions have higher FPR than the northern at a given temperature. Also, at -30 oC, the southern polar regions have much lower FPR than the northern, which contradicts with the two referred papers. 4) Line 255, I do not see any downdrafts (negative values I guess) in the figure. 5) Line 258, again, it is hard to understand why updraft positively correlated to the occurrence of ice clouds at -20 oC. The referred paper is looking at large scale motion impacts on supercooled liquid fraction. Generally strong updrafts have higher supersaturation and lead to the more liquid formation, and downdrafts cause liquid layer to evaporate, and thus impact the supercooled liquid fraction, which makes sense. I have difficulty to figure out the physical process that updraft velocity lead to the occurrence of ice cloud at -20 oC given that deposition ice nucleation is rare in the atmosphere.

Figure 8 and Line 390: 1) RH has strong impacts on ice nucleation only for the deposition mode. While at the temperatures between -40 and 0 oC, ice particles are always formed from the freezing of liquid droplets in the atmosphere. Therefore, I do not think it is necessary to include the discussion of RH relationship with dust. In another word, large-scale motion has strong impacts on cloud thermodynamic phase partition, mainly

on the formation and evaporation of liquid droplets, but it has little impacts on ice nucleation at the heterogeneous temperature range; 2) Please explain why dust aerosol mixing-ratio increases with isotherm height; 3) Why there are not downdraft?

Minor comments:

"CLOUDAT" -> CloudSat.

Line 120: "obtaining good results". How good are the simulated dust mixing-ratios compared with observations?

Line 125: Jan'0 -> Jan'09

Line 138 and 140: the word 'pixel' is confusing. How does pixel be defined in this study? Is this a layer bin in each lidar or radar profile? Then how to know whether a pixel is precipitating or not?

Line 164: how is cloud volume fraction derived? What's the benefit using this cloud volume fraction comparing FPR?

Line 182 and equation 3.1: How is equation 3.1 is derived or why does this definition differentiate the clear sky and liquid phase condition?

Line 241: It is not a good starting sentence, and should be put behind Line 243.

Line 320: Any guess of the common mechanism?

Figure 6: Given similar dust mixing-ratio at -15 oC, the ice occurrence is ∼10% higher at mid-latitudes than polar regions, what's the reason?

Figure 7: at -30 oC, ice occurrence shows little correlations with dust, but rather has dramatic latitude differences. Any speculations of the reason? I guess at such colder temperatures, other aerosol particles such as soot are also good INPs. It might be interesting to also look at the relationships between FRP and soot particles.

References: Kanitz,ÂăT.,ÂăP. Seifert,ÂăA. Ansmann,ÂăR. Engelmann,ÂăD. Althausen, C. Casiccia, and E. G. Rohwer (2011), Contrasting the impact of aerosols at northern and southern midlatitudes on heterogeneous ice formation, Geophys. Res. Lett., 38, L17802, doi: 10.1029/2011GL048532.

Interactive
comment

---

## Referee Comment (RC4) · Anonymous Referee #4 · 12 Dec 2018

This paper shows how satellite observations can be associated with aerosol model re-analysis to infer the impact of dust on cloud thermodynamic phase transition. The authors used LIDAR and RADAR measurements from the A-train to retrieve information on the cloud thermodynamic phase using different products (DARDAR, GOCP) and the reanalysis MACC to co-locate dust mixing ratio and updraft. Therefore, the study retrieves the frequency phase ratio as a function of dust mixing ratio constrained for different regimes of latitude, temperature, season, etc. The aerosol-cloud interaction problem is a difficult subject to study with observations because aerosols and clouds properties cannot be spatially and temporally co-located. The use of satellite and re-analysis circumvents the problem. The results show that the cloud ice fraction increases

with the concentration of dust, suggesting that dust plumes contain ice nuclei which is line with previous studies as mentioned by the authors. Each effects are quantified, the manuscript is well written and well structured in my opinion. The objectives of the study are clearly mentioned in the introduction and the results associated with their significances are stated in the conclusion.

The topic of this study matches very well with the journal topics and is of great interest for the atmosphere community, in particular people studying aerosol-cloud interaction. However, I consider that the article needs important revision that I estimate necessary to be published: the discussion on meteorological parameter impacts is too short, the data section is too short... (see below for detailed descriptions).

Major revisions: 1. Meteorological parameters have a larger impact on cloud properties than aerosols (Gryspeerdt et al., 2016). Different meteorological regimes can change the aerosol-cloud interaction by an order of magnitude. Even if you mention in your paper the meteorological parameters (sections 4 and 5), it is missing in the paper. You refer to humidity, but the stability is also an important parameter in the aerosol-cloud interaction. The spatial resolution of ERA can seem coarse but it could constrain your situation and could avoid any correlation you are referring to (line. 400): You might not have the atmospheric state at the cloud but it refers to general atmospheric processes which are important as well.

Also, the boxes you considered based on latitudes-longitudes contain both land and ocean which are in different regimes of aerosols and meteorological parameters, I would like to see a differentiation between land and ocean. Moreover, you based your study on dust aerosols, but other parameters can have an impact on the ice fraction (soot, sea salt, sulphate...), the low correlation you observe in the hemisphere north could also be due to the fact that there are more different aerosol types which can act as IN as well.

2. You refer to the maximum number of points you can retrieve in line 126-127, but the

actual number of points never appears in the article, I suggest to add the number of data points in the Figures 6, 7, 8 for each point. The work is based on statistical analysis so the number of points is important, especially if you compare different regimes. I am particularly concerned by Figure 4-b and the increase of ice occurrence for latitudes lower than 73S (maybe the results are not statistically significant because you do not have enough data points).

Also I am concern by Equation 3.6: If I understood, you average for each dust mixing ratio bin to have constant number of 10 data points. This method artificially increases the correlation coefficient. Could you measure the Pearson correlation based on the 2-D histograms. For example, in Figure 6-a, what would the correlation coefficient be if you consider all the couples (iceOccurence-FineDustMixingRation) without averaging for Fine dustmixing-ration bins first.

3. The data section lacks necessary information. The satellite needs to be described more precisely with information about the performance of the algorithms: When they are compared to in-situ or ground-based measurements, how do they perform? What are the methods to derived cloud properties? The same goes for MACC, in line 120 you refer to "good results", can you develop and quantify.

4. You plotted the uncertainties in your figure but you do not refer to them in your text. For example in line 293, you use "notably higher", but if you consider the uncertainties in the figure, the difference is not that high. Can you comment on that?

Minor revisions:

- Introduction: There is plenty of different methods to study the aerosol-cloud interactions. The method you are using present fair advantages. A paragraph is needed to highlight this.

- The use of "e.g." needs a coma, example in line 28: (e.g., Patagonia, South Africa, and Australia)

- Method section: Are clouds vertically co-located with the dust mixing ratio?

- Figure 1 is not described, you need to introduce it and describe it.

- Figure 8-a and 8-b are not presented, can you describe them before referencing to them?

- Line 179: Why do you use 30 degree? Is there any specific reason? Did you try with boxes of 20 degree for example? You refer to "optimize the number of different satellite swaths", I do not understand, can you develop?

- Equation 3.1: Has it been used in a previous study?

- Line 226: "2,5" should be "2.5"

- Line 278: How do you explain that you have a larger correlation in the southern hemisphere compare to the northern hemisphere? Can you speculate?

- Line 309 - 315: This paragraph is not clear, can you reformulate?

- Line 345 : "In contrast ..." : For the other cases where you have a lower correlation, can it mean that the glaciation happened before, and therefore you do not find a good correlation but dust plumes still contain IN, can you comment on that?

- Line 437: you mention in the paper that you are substituting the m2/3 to do a linear regression, but not in the conclusion, so it is confusing when you refer to m2/3 as linear. Can you clarify this in the conclusion?

- Figure 2 caption: "— are reclassified a ice", I think you mean liquid.

- Figure 2: You put arrows on the colorbars but it cannot be greater than 100% ice, or 100% liquid.

- Table1: It took me a while to understand Table 1, there is a lot of information, and some of them are never mentioned in the text. Can you simplify it? I have the feeling that it could actually be two different tables.

References: Gryspeerdt, E., Quaas, J., & Bellouin, N. (2016). Constraining the aerosol influence on cloud fraction. Journal of Geophysical Research: Atmospheres, 121(7), 3566-3583.

---

## Author Comment (AC1) · 29 Jan 2019

General Comments: This paper explores important relationships between aerosol properties (specifically mineral dust loading), cloud phase, and other dynamic and thermodynamic variables. I commend the authors for attempting to provide a global perspective on this challenging to measure and understand phenomenon. The satellite-based perspective on aerosol cloud interactions can provide more insight into the global extent of aerosol influences than obtainable from measurements from a single observatory. In general, I think that the analysis completed is interesting and potentially sheds some light onto the relationships between mineral dust and cloud properties. Having said that, I do have some reservations and questions about the manuscript, and therefore would like to see some additional work completed before this paper can be accepted for publication. Additionally, I found the manuscript to be very dense, and found myself having to reread sections on a regular basis. I'm not sure whether this was the result of the frequent use of abbreviations, or the writing style, or something else. However, it was a challenge to read, which is unfortunate given the science that is in the paper.

We sincerely thank Anonymous Referee #1 for pointing out several potential sources of uncertainty in our methods. We have modified several parts of the paper to address these points.

Specific Comments:
1.a) I wonder to what extent the different products used in this evaluation are consistent with one another. This is particularly a large question for the reanalysis derived estimates of temperature, humidity and vertical velocity. Since the reanalyses rely on models to provide input on clouds and dynamics, the values represented in these products must be internally consistent — however, this may not always match the real atmosphere. This is particularly true in areas of cloud cover, where the model clouds and real clouds may not match in time, location and phase.

We are aware of this issue and addressed the problem in lines 371-373 referring to the simulated dust mixing-ratio. Similarly, we assume that *statistically* the derived estimates of temperature, humidity and vertical velocity are accurate enough to evaluate their behaviour at different day-to-day loadings of mineral dust.

We agree that the comparison between the cloud phase from reanalyses and satellite data would produce a significant source of uncertainty. Therefore, we had already excluded the estimates of ice occurrence from the reanalysis to avoid issues resulting from such miss-colocations.

Nevertheless, we have now extended lines 371-373 to include this issue: "Similarly, the atmospheric parameters from the reanalysis may not match the exact position of the clouds in the satellite retrievals. However, we expect the atmospheric parameters to match in average the large-scale conditions influencing the observed clouds."

1.b) Therefore, I wonder whether there may be instances where the model (reanalysis)-produced thermodynamic state is inconsistent with the clouds detected from satellite measurements, potentially biasing the evaluation of observed cloud phase into different temperature regimes from reality. Some discussion on the potential for this to occur would be helpful.

It is possible to find cases where reanalysis and detected clouds have different temperatures. Although this may indeed contribute to the overall errors, we do not believe that this may present a major bias towards a certain cloud phase. This is because the temperature profiles used to bin the measurements and reanalysis into the different temperature regimes (3 K bins) are independent to each other. Therefore, a systematic bias would have been noticed in the comparison between both cloud phase products.

We have now added this point to the Discussion.

2.a) There is a substantial question related to the ability to truly detect relationships between dust and cloud phase in the absence of sufficient constraints on environmental state and dynamics. The co-variability that is demonstrated is interesting, but how can one be sure that this is the result of the aerosol, and not of the dynamical forcing on the cloud?

We agree that the constrains on the data are not enough to prove a causal relationship between dust aerosol and cloud phase. Given the limited data available, we found that additional temporal constrains on the dataset (e.g. dynamic regimes) leaded to insufficient sample sizes in the individual regimes. Therefore, we focused in analysing the co-variability between dust aerosol and some key dynamical parameters in Fig 8 (RH, cloud height and updraft) to rule out the possibility of some of these parameters driving the co-variability observed in Fig 6. From this analysis, we concluded that the co-variability between dust mixing-ratio variability and dynamics appears to be too low to support that the observed increases of ice occurrence in Fig. 6 are controlled by dynamics.

However, we agree that a methodology to isolate the effect of aerosols from the correlated effect of dynamics is still lacking. Therefore, we have now emphasized the need for such a methodology in our outlook: "Additionally, the further development of a methodology to isolate aerosol-cloud interactions from atmospheric dynamics has the potential to reduce much of the uncertainty found in this study."

See also the response to Anonymous Referee #4: "We agree that stability is a useful parameter for separating between aerosol-cloud interaction regimes.
However, #4.1a shows that we found no significant difference in the day-to-day correlation between dust and ice occurrence for different stability regimes (defined as "unstable", "neutral" and "stable"). Nor were the day-to-day changes in stability associated with changes in ice occurrence.

We used lower-tropospheric static stability (LTSS) following Li et al. 2017 (defined in (Klein and Hartmann, 1993))."

2.b) Particularly at high latitudes, where figure 8 appears to show a relationship between dust mixing ratios and vertical velocities, it could be challenging to discern the impacts of the dynamics from those of the aerosols. I will say that it seems to me that the relationship is such that you would not necessarily expect that the dynamics and aerosols would work in the same direction – with increasing mineral dust loading you have increasing vertical velocities (upward), which would support enhanced supersaturation. Therefore, the cloud liquid would increase in response to the updraft but decrease in response to the mineral dust loading. Assuming I have that correct, this could help to support the idea that the changes are the result of aerosols and not dynamics, but I think that a good amount of discussion on this topic is warranted.

We thank Anonymous Referee #1 for this helpful argument. The correlation between dust mixing-ratio and updraft suggest indeed that (everything else held constant), the ice occurrence should decrease at higher dust mixing-ratios. In fact, larger updrafts favour supersaturation and therefore CCN activation, droplet growth and inhibition of the WBF (Wegener–Bergeron–Findeisen) process. All three processes would lead to a lower cloud ice occurrence. One could then argue that the increase in ice occurrence for higher dust mixing-ratio should be even larger if the effect of updrafts is considered. We agree however, that this would raise a larger discussion. Immersion freezing, for example, requires a saturation over liquid water. This could result in updrafts promoting heterogeneous nucleation.

The spatial correlations in the study of J. Li et al. 2017 show actually an increase in cloud ice occurrence for higher large-scale updrafts. Although not included in the paper, we also found a day-to-day increase in cloud ice occurrence for higher updrafts. Therefore, the in-depth analysis of the dust-updraft-iceOccurrence co-variability would need a new evaluation of the relationship between large-scale updraft and ice cloud occurrence. As mentioned by Anonymous Referee #1, this would lead to a large (and necessary) discussion but this is outside the scope of this study.

This issue has been now commented in the Discussion.

• Line 129: What, if any, sensitivity is there to the order of the averaging?

Temperature is the dimension along which the cloud phase variability is highest. If temperature were the first ordering dimension, the information about each column would be biased towards the temperatures with the largest cloud cover. For example, a larger cloud fraction at higher temperatures would lead to a lower average ice cloud occurrence. This is because many columns would only contain data for higher temperatures (lower ice cloud occurrence).

Figure #1. 2.b shows the effect of reversing the averaging order. In the northern high latitudes (cyan box), the cloud volume fraction is significantly higher at higher temperatures (See S8 in the supplement). This produces a bias towards lower ice occurrences when averaging temperature at first.

From sect 4.2 and 4.3 we established that temperature, followed by latitude are the dimension with the largest variability of ice cloud occurrence. By averaging such dimensions last, we minimize the bias produced while averaging a dataset containing missing values.

[Figure]

Figure #1.2.b *Effect of averaging order for FPR$_{GOCCP}$*

[Figure]

S8. Zonal mean of cloud volume fraction [%]. CALIOP-GOCCP 2007-2010.

• Lines 139-140: Please describe what cloud depth constitutes precipitating vs. nonprecipitating in 2B-CLDCLASS

As briefly mentioned in lines 88-89, the 2B-CLDCLASS product uses mainly the radar reflectivity to classify clouds as "precipitating". The radar is sensitive to large particles (e.g., rain drops) and therefore clouds with a reflectivity larger than a given temperature-dependent threshold are defined as "precipitating". The fifth range gate (~1.2 km above ground level) is used for the classification. The threshold is defined between -10 and 0 dBZ for temperatures between -10 °C and 0 °C and constant outside this temperature range (Hudak et al., 2009).

Hudak et al. 2009 offers a validation of the 2B-CLDCLASS precipitation product and a brief description (paragraphs 10-11) of the precipitation algorithm.

References: Hudak, D., Rodriguez, P. and Donaldson, N.: Validation of the CloudSat precipitation occurrence algorithm using the Canadian C band radar network, J. Geophys. Res. Atmos., doi:10.1029/2008JD009992, 2009.

These points have been added to the Methods section.

• Line 179: 30x1.825 deg — depending on latitude, this can be a very large amount of area. Therefore, I would be concerned about sub-grid variability, particularly at lower latitudes. For example, dust and cloud could be on different sides of a front within a given grid box. This is discussed a bit starting on line 370. Additionally, there could be significant gradients in phase over such a large horizontal extent, particularly around mid-latitude frontal zones, coastlines, etc. Such gradients may have little to do with the dust, but everything to do with changes in forcing.

As mentioned by Anonymous Referee #1, our approach does not consider such post-/pre-frontal differences in the analysis. Whether the relationship shown in our study is also applicable within mesoscale convection systems is a very interesting question.

Nevertheless, we believe that for studying such mesoscale phenomena another toolset is required, including mesoscale modelling and tracking of frontal systems.

• Figures 3 and 4: Recommend removing the dotted vertical lines to improve clarity.
Removed.

• Figure 3: "Error bars" — this isn't really error, is it? Just the standard deviation and variability?
Renamed.

• Figure 4: Same comment about "error bars"
Renamed.

---

## Author Comment (AC2) · 29 Jan 2019

General comments: The authors analysed the impact of fine and coarse dust aerosols on the day-to-day variability of stratiform cloud glaciation occurrence by using 4 years' satellite cloud phase products and MACC aerosol reanalysis. Compared to the previous study, this study mainly focused on the day-to-day variability of cloud phase. The results showed that the phases of stratiform clouds is highly dependent on temperature and latitude, and dust aerosol mixing ratio is anti-correlated with the average occurrence of fully glaciated stratiform clouds. Generally speaking, the paper is interesting, and tables and graphics are well constructed. As a result, I am recommending the paper be accepted with major revisions if the authors response properly my comments. Some main questions and comments I have are listed below in the specific comments to the authors.

We thank Anonymous Referee #2 for his encouraging comments.

Specific Comments:

(1.a) One of my concerns is: The applicability of MACC aerosol reanalysis, especially over the Southern Ocean, where ground-based measurements are sparse. There is no direct observation evidence of dust aerosol to prove the applicability of MACC aerosol reanalysis.

Indeed, all evidence is from northern hemisphere. However, until such studies are available in the southern hemisphere, we assume that the positive past evaluations in the northern hemisphere reflect that the emission, transport and deposition mechanisms should offer a valid estimate in the southern hemisphere.

We have emphasized in the outlook the need of future validations of the MACC aerosol reanalysis in the southern hemisphere.

See also response to Anonymous Referee #3: *"...showing a mean bias of 25% between MACC and LIVAS (dust product based on CALIPSO satellite) over Europe, northern Africa and Middle East. Additionally, the correlation between MACC and AERONET (network of ground-based remote sensing stations) was found in the range of 0.6 over the Sahara and Sahel to 0.8 over dust transport regions."*

(1.b) In addition, the variable used in this study is dust mixing-ratio? Isn't mass concentration? I think that the dust mass concentration should be more proper for this analysis.

The height of the studied isotherms varies from 0 up to 8 km from pole to the equator (See supplement S12). Because the different atmospheric pressure at these levels lead to different molar volumes of air, mixing-ratio (kg/kg) has the advantage of being independent of pressure. However, we believe that both mixing-ratio and dust mass concentrations can be valid parameters. Additionally, mixing-ratio (kg/kg) is the original output of the MACC reanalysis.

(2) One suggestion: The section 3 includes too much information, and it is easy to confuse the readers. May you provide us one table or flow chart? The authors also reconstruct this section to make the method clearer. For example, move the lines 177-195 to the first paragraph, and following sentences interpret these variables.

We have now included a flow chart as suggested. The Methods section has been also reordered to match the diagram.

[Figure]

Figure 1. Flow chart of the data processing steps.

(3) Line 125: What is the meaning of the Jan'0?
Jan'09 (January 2009). Corrected and clarified.

(4) Line 140: Please provide the detailed information about the classification of nonprecipitation.

See response to Anonymous Referee #1: "As briefly mentioned in lines 88-89, the 2B-CLDCLASS product uses mainly the radar reflectivity to classify clouds as "precipitating". The radar is sensitive to large particles (e.g., rain drops) and therefore clouds with a reflectivity larger than a given temperature-dependent threshold are defined as "precipitating". The fifth range gate (~1.2 km above ground level) is used for the classification. The threshold is defined between -10 and 0 dBZ for temperatures between -10 °C and 0 °C and constant outside this temperature range (Hudak et al., 2009).

Hudak et al. 2009 offers a validation of the 2B-CLDCLASS precipitation product and a brief description (paragraphs 10-11) of the precipitation algorithm.

References: Hudak, D., Rodriguez, P. and Donaldson, N.: Validation of the CloudSat precipitation occurrence algorithm using the Canadian C band radar network, J. Geophys. Res. Atmos., doi:10.1029/2008JD009992, 2009.

These points have been added to the Methods section."

(5) For the equations 3.2-3.5, FPR or FPR*?
FPR is right. FPR* (Equation 3.1) is only used in Sect 4.1 as stated in text. FPR* is used exclusively to ease the visualization of the thermodynamic phase in the case study (to show only clouds with significant cover).

This has been now clarified in the methods section.

See also the response to Anonymous Referee #3: *"The sole purpose of the definition of FPR\* is to plot the cloud phase as a range between LIQUID and ICE while also blending out non-significant clouds (of cvf~0). Here the cloud volume fraction cvf is used merely as a filter to aid the visualization in Fig. 2."*

(6) The main concern is: the authors how to peel off the impact of meteorological condition from dust loading. Because the aerosol and dynamical factor usually are co-variability. Thus, some discussions about dynamical factor are necessary.

See response to Anonymous Referee #1: *"We agree that the constrains on the data are not enough to prove a causal relationship between dust aerosol and cloud phase. Given the limited data available, we found that additional temporal constrains on the dataset (e.g. dynamic regimes) leaded to insufficient sample sizes in the individual regimes. Therefore, we focused in analysing the co-variability between dust aerosol and some key dynamical parameters in Fig 8 (RH, cloud height and updraft) to rule out the possibility of some of these parameters driving the co-variability observed in Fig 6. From this analysis, we concluded that the co-variability between dust mixing-ratio variability and dynamics appears to be too low to support that the observed increases of ice occurrence in Fig. 6 are controlled by dynamics.*

*However, we agree that a methodology to isolate the effect of aerosols from the correlated effect of dynamics is still lacking. Therefore, we have now emphasized the need for such a methodology in our outlook: "Additionally, the further development of a methodology to isolate aerosol-cloud interactions from atmospheric dynamics has the potential to lift much of the uncertainty found in this study."*

*(...)*

*The correlation between dust mixing-ratio and updraft suggest indeed that (everything else held constant), the ice occurrence should decrease at higher dust mixing-ratios. In fact, larger updrafts favour supersaturation and therefore CCN activation, droplet growth and inhibition of the WBF (Wegener–Bergeron–Findeisen) process. All three process would lead to a lower cloud ice occurrence. One could then argument that the increase in ice occurrence for higher dust mixing-ratio should be even larger if the effect of updrafts is considered. We agree however, that this would rise a larger discussion. Immersion freezing, for example, requires saturation over liquid water. This could result in updrafts promoting heterogeneous nucleation.*

*The spatial correlations in the study of J. Li et al. 2017 show actually an increase in cloud ice occurrence for higher large-scale updrafts. Although not included in the paper, we also found a day-to-day increase in cloud ice occurrence for higher updrafts. Therefore, the in-depth analysis of the dust-updraft-iceOcurrence co-variability would need a new evaluation of the relationship between large-scale updraft and ice cloud occurrence. As mentioned by*

*Anonymous Referee #1, this would lead to a large (and necessary) discussion but we think that it lays outside the scope of this study.*

*This issue has been now commented in the Discussion."*

And the response to Anonymous Referee #4 :" *We agree that stability is a useful parameter for separating between aerosol-cloud interaction regimes. However, figure #4.1a shows that we found no significant difference in the day-to-day correlation between dust and ice occurrence for different stability regimes (defined as "unstable", "neutral" and "stable"). Nor were the day-to-day changes in stability associated with changes in ice occurrence. We used lower-tropospheric static stability (LTSS) following Li et al. 2017 (defined in (Klein and Hartmann, 1993))."*

---

## Author Comment (AC3) · 29 Jan 2019

General comments: The manuscript presents an interesting study of mineral dust impacts on stratiform cloud glaciation occurrence using satellite remote sensing measurements and MACC aerosol reanalysis daily product. Mineral dust is one of the major atmospheric aerosols and are also efficient INPs. However, it is still challenging to quantify their impacts on ice nucleation in clouds and cloud thermodynamic phase partitioning. Their findings in this study confirm that mineral dust has positive correlations with ice cloud occurrence on a global scale even after limiting other impact factors. The approach presented in this study is very unique. Personally, I like the approach and also am very happy with their findings. However, many of their discussions are hard to explain the observed features (sometimes contradict with previous studies). Besides, the manuscript is not well organized and poorly written (especially the comparisons between FPRDARDAR and FPRGOCCP) and therefore is hard to read. Overall, I believe serious revisions are needed before the manuscript be accepted by ACP.

We thank Anonymous Referee #3 for his constructive comments. We realize that the discussion about our Figures was lacking some important information. We have addressed all points and believe that a significant improvement has been made.

**Major comments:**

**Fig1.:**

**1) About the sample size, what causes the missing datapoints?**

There are various sources of missing data:

- a) The satellite swaths (orbits) produce different density of data at different latitudes.
- b) Using only night-time data the sample size in the meteorological summer time (shorter nights) is lower.
- c) The cloud phase data is (for obvious reasons) less frequently available in areas and heights (temperatures) of low cloud cover.
- d) Certain temperature ranges near the poles can lay below the surface temperature and therefore no information is available for such temperatures (specially in the Antarctica).

These points have been now included in the Methods section.

See also Supplement S14 for an in-depth view of the distribution of the sample size across seasons, temperature and regions.

**2) Why do the high-latitudes have less datapoints than the mid-latitudes?**

See last question.

**3) Is the sample size same or similar for DARDAR and MACC?**

The MACC reanalysis dataset does not have missing values as it is produced from a model output. The DARDAR product uses only satellite swaths were both CALIPSO and CLOUDSAT satellite retrievals are available. Additionally, in both datasets we have omitted pixels where the CALIOP signal is fully attenuated. Therefore, the DARDAR sample size is equivalent to the GOCCP excepting downtimes of the CPR (the most relevant being 07/12/2009 - 16/01/2010) and other minor exceptions.

Fig 2. and related discussion:

1) According to line 143-144 and line 153, "lidar-not-fully-attenuated" and "only ice pixels are considered as ice clouds", I would expect basically DARDAR and GOCCP have very similar FPR. For example, in Fig 2., DARDAR detected the ice virga below the liquid layer top. However, generally, the liquid layer at the top will fully attenuate the lidar signal (also as stated in line 119). Therefore, given the filters listed in line 143-144, I do not understand why DARDAR still show an ice layer below;

The clouds between -40°N and -45°N and between -6°C and -33°C are classified as "altostratus" by the 2B-CLDCLASS product. In most cases, such stratiform clouds are thin enough to be penetrated by the lidar.

FPR\_GOCCP: The detected ice virgae below the liquid cloud top suggests the cloud top did not fully attenuate the lidar signal (cloud not thick enough). The number and/or size of the ice particles near the cloud top probably was not enough to increase the depolarization ratio above the threshold value for the GOCCP algorithm, therefore was classified as liquid.

FPR\_DARDAR: In the decision tree of the DARDAR algorithm there are multiple alternatives for a mixture of cloud droplets and ice particles (e.g. at cloud top) to be classified as ice only (Mioche et al., 2014):

- a) If the lidar backscatter signal ( $\beta$ ) is lower than 2.10-5 m-1 sr-1
- b) If not "a)": If it is "weekly attenuated" (less than 10 times) or "not rapidly attenuated" (at a depth larger than 480 m).
- c) If not "b)": If the vertical thickness of the cloud is larger than 300 m (equivalent to 5 pixels with a lidar resolution of 60 m).

Therefore, there are plenty cases where a mixed-phase cloud (and specially optically thin stratiform clouds) can be mis-classified as ice only in the DARDAR product and consequently in the FPR\_DARDAR variable. In this specific case, we speculate that "c)" is the most probable cause because of the large vertical extension of the cloud around (1 to 3km using a lapse rate of -10°C/km for the estimation).

FPR\_ALT\_DARDAR: In the case of droplets and ice particles coexisting at cloud top, we expect that at some location of the swath, the cloud droplets will be enough in number to be classified as liquid (strong attenuation) by the DARDAR algorithm. If this is the case, the entire gridbox value of FPR\_ALT\_DARDAR will be LIQUID (interpreted as a non-completely glaciated cloud).

Further explanation: The GOCCP algorithm is unable to detect ice in mixed-phase clouds and the DARDAR algorithm tends to classify mixed-phase clouds as ice. Therefore, we avoid using the "frequency of cloud ice" (FPR) to compare the GOCCP and DARDAR products. Instead, we use a parameter which considers the limitations of both products. In FPR\_ALTDARDAR, mixed-phase clouds that would be otherwise classified as ice are now classified as liquid. This "recreates" the inability of the GOCCP algorithm to detect ice in mixed-phase clouds. Therefore, the "frequency of completely glaciated clouds" (FPR\_ALT) allows a comparison between both algorithms (FPR\_ALTDARDAR and FPRGOCCP).

We have added this explanation to the Sect. 4.1.

2) Comparing 2a, b, d, e, and f there are not clouds at latitudes between -40 and -45 and temperatures between -6 and 3, there shows no cloud in 2a, b, and e, but does have FPRDARDAR and FPRGOCCP in 2d and f, what's the reason for the differences? In this gridbox, there is (erroneously) a small cloud fraction corresponding to a cumulus nimbus cloud (See figure #3.2).

The FPR\_ALT\_DARDAR is 50% (50/50 ice and liquid) which is plotted white in 2.e. This is the result of the satellite orbit passing two adjacent gridboxes at the same latitude. One of this gridboxes was classified as liquid and the other as ice.

The FPR plots shown in Figure 2 were taken directly from the raw data and equivocally included the phase and cloud fraction of non-stratiform clouds (which were thought negligible for the case study).

We apologize for the mistake. This has been now corrected.

Figure #3. 2 Case study

Figure 2 (Modified)

3) Comparing 2d and e, at latitudes between -40 and -45, the whole profiles shows ice cloud in 2d but it is liquid in figure 2e. Even with all the rules described in line 157-158, it is still difficult to understand why it is liquid in figure 2e (the grid box is still 1.875 x 1.875, right?)

The grid box is still  $1.875 \times 1.875$ . See the answer to question. The rules have been now clarified as discussed in the previous answer.

**Fig.3:**

1) I really object to put dust mixing ratio in this figure. There is no physical relationship between dust mixing-ratio and temperature. Dust mixing ratio decreases with height near the source region and temperature generally decrease with height in the atmosphere. But physically, there is no relationship between temperature and dust!

We agree that it may be disorienting to plot the dust mixing-ratio together with the ice occurrence.

We have changed the statement "Of course, this results alone from the relationship between temperature and dust, and it is therefore a good example of a correlation (between dust and FPR) without a direct causality." (lines 237-239), to: "The purpose of plotting both variables together is not to suggest a physical relationship, but to give an insight of the trend of both parameters along temperature."

2) In line 235, "mixing-ratios are higher near the surface than at high altitudes", this is not true for transported dust layer such as over the high latitude regions.

This is a valid exception. However, he intention of Fig. 3 is only to explain the mean vertical distribution of dust at different temperatures.

3) Overall, this study is trying to look at the relationship between pure ice cloud fractions and dust mixing ratios. Therefore, comparing FPRDARDAR and FPRGOCCP has little contribution to the main research topic in this study and the descriptions of converting different FPRs are quite confusing and distracting. In fact, in the proceeding sections, only FPRGOCCP is used most of the time. Therefore, I would suggest just use FPRGOCCP in the study.

We agree that this would simplify the study. However, we believed that the use of both products (FPR\_ALTDARDAR and FPRGOCCP) adds confidence to the results shown in Table 1 and rules out any artefacts arising from different cloud phase classification algorithms.

**Fig. 4:**

1) It is surprising that 'maximum of FPR is located near the Equator'. Kanitz et al. (2011) showed that Northern mid-latitudes have much higher ice-containing cloud fraction than the tropical regions. Do the authors have speculations why the results are so different?

This is a reflection of the larger fraction of cirrus clouds near the equator. Indeed, between 0 and -40°C cirrus clouds are almost exclusively found near the equator, as the polar cirrus clouds occur at a much lower temperature. The results from Kanitz et al. 2011, corresponds to a 2-month local measurement in Cape Verde (January- February 2008, see Ansmann et al., 2009) and are therefore not a suitable comparison to the climatological FPR presented here. These results are consistent with the ones reported in J. Li et al. 2017 (See #4. 1.c). Additionally, it was shown in Fig. 7.19 of the dissertation work of Seifert (2010), that the

majority of the altocumulus layers observed over Cape Verde formed in the absence of mineral dust at height levels where westerly upper-tropospheric winds dominate.

References:

Ansmann, A., Tesche, M., Seifert, P., Althausen, D., Engelmann, R., Fruntke, J., ... Müller, D. (2009). Evolution of the ice phase in tropical altocumulus: SAMUM lidar observations over Cape Verde. Journal of Geophysical Research Atmospheres. https://doi.org/10.1029/2008JD011659

http://nbn-resolving.de/urn:nbn:de:bsz:15-qucosa-71167

---

## Author Comment (AC4) · 29 Jan 2019

This paper shows how satellite observations can be associated with aerosol model reanalysis to infer the impact of dust on cloud thermodynamic phase transition. The authors used LIDAR and RADAR measurements from the A-train to retrieve information on the cloud thermodynamic phase using different products (DARDAR, GOCCP) and the reanalysis MACC to co-locate dust mixing ratio and updraft. Therefore, the study retrieves the frequency phase ratio as a function of dust mixing ratio constrained for different regimes of latitude, temperature, season, etc. The aerosol-cloud interaction problem is a difficult subject to study with observations because aerosols and clouds properties cannot be spatially and temporally co-located. The use of satellite and reanalysis circumvents the problem. The results show that the cloud ice fraction increases with the concentration of dust, suggesting that dust plumes contain ice nuclei which is line with previous studies as mentioned by the authors. Each effects are quantified, the manuscript is well written and well structured in my opinion. The objectives of the study are clearly mentioned in the introduction and the results associated with their significances are stated in the conclusion. The topic of this study matches very well with the journal topics and is of great interest for the atmosphere community, in particular people studying aerosol-cloud interaction. However, I consider that the article needs important revision that I estimate necessary to be published: the discussion on meteorological parameter impacts is too short, the data section is too short... (see below for detailed descriptions).

We thank Anonymous Referee #4 for the new insights and ideas. We have carefully reviewed all concerns and made some new analysis of the data. We believe that the overall quality of the manuscript has been now greatly improved.

**Major revisions:**

1.a Meteorological parameters have a larger impact on cloud properties than aerosols (Gryspeerdt et al., 2016). Different meteorological regimes can change the aerosol-cloud interaction by an order of magnitude. Even if you mention in your paper the meteorological parameters (sections 4 and 5), it is missing in the paper. You refer to humidity, but the stability is also an important parameter in the aerosol-cloud interaction. The spatial resolution of ERA can seem coarse but it could constrain your situation and could avoid any correlation you are referring to (line. 400): You might not have the atmospheric state at the cloud but it refers to general atmospheric processes which are important as well.

We agree that stability is a useful parameter for separating between aerosol-cloud interaction regimes. However, Figure #4.1a shows that there is no significant difference in the day-to-day correlation between dust and ice occurrence for different stability regimes (defined as "unstable", "neutral" and "stable"). Nor were the day-to-day changes in stability associated with changes in ice occurrence. For the analysis we used the lower-tropospheric static stability (LTSS) defined in Klein and Hartmann, 1993 and following Li et al. 2017.

Figure #4. 1.a : FPRGOCCP vs dust mixingratio for different stability regimes.

Lower-tropospheric static stability (LTSS)

1.b Also, the boxes you considered based on latitudes-longitudes contain both land and ocean which are in different regimes of aerosols and meteorological parameters, I would like to see a differentiation between land and ocean.

Figure #4.1b shows this differentiation. The shift between the curves results mostly from the differences in dust mixing-ratio between sea and land. We note that we find no significant changes in cloud occurrence between land and sea for temperatures ranging from -10 °C to -40 °C. This partly contradicts the results from Tan et al. 2014.